# Learning More with Less: A Dynamic Dual-Level Down-Sampling Framework for Efficient Policy Optimization

Chao Wang[1][*][†]   Tao Yang[2][*][‡]   Hongtao Tian[2]   Yunsheng Shi[2]

Qiyao Ma[3][†]   Xiaotao Liu[2]   Ting Yao[2]   Wenbo Ding[1][‡]

[1]Tsinghua University    [2]WeChat, Tencent    [3]University of California, Davis

## Abstract

Critic-free methods like GRPO reduce memory demands by estimating advantages from multiple rollouts but tend to converge slowly, as critical learning signals are diluted by an abundance of uninformative samples and tokens. To tackle this challenge, we propose the **Dynamic Dual-Level Down-Sampling ($D^3S$)** framework that prioritizes the most informative samples and tokens across groups to improve the efficiency of policy optimization. $D^3S$ operates along two levels: (1) the sample-level, which selects a subset of rollouts to maximize advantage variance ($\text{Var}(A)$). We theoretically proved that this selection is positively correlated with the upper bound of the policy gradient norms, yielding higher policy gradients. (2) the token-level, which prioritizes tokens with a high product of advantage magnitude and policy entropy ($|A_{i,t}| \times H_{i,t}$), focusing updates on tokens where the policy is both uncertain and impactful. Moreover, to prevent overfitting to high-signal data, $D^3S$ employs a dynamic down-sampling schedule inspired by curriculum learning. This schedule starts with aggressive down-sampling to accelerate early learning and gradually relaxes to promote robust generalization. Extensive experiments on Qwen2.5 and Llama3.1 demonstrate that integrating $D^3S$ into advanced RL algorithms achieves state-of-the-art performance with generalization while requiring *fewer* samples and tokens across diverse reasoning benchmarks.

## 1 Introduction

Reinforcement Learning (RL) has become instrumental in aligning Large Language Models (LLMs) with human values and preferences (Ouyang et al., 2022), leading to the emergence of various alignment algorithms. Among these, critic-free methods like Group Relative Policy Optimization (GRPO) (Shao et al., 2024) and Group Sequence Policy Optimization (GSPO) (Zheng et al., 2025) have marked a crucial step towards greater memory efficiency. These methods estimate advantages using relative rewards from a group of sampled responses, thereby eliminating the need for a separate critic network (Shao et al., 2024). However, while the memory bottleneck is alleviated, efficiency challenges remain. The precision of estimation of advantages utilized in training depends on the quality of the sampled groups. Larger groups risk diluting critical learning signals from a few key samples and tokens, as these signals can be overshadowed by the averaging effect of numerous *undifferentiated* samples (e.g., groups dominated by mostly correct or incorrect samples in mathematical reasoning tasks). Conversely, smaller groups may struggle to yield diverse samples due to insufficient sampling. This trade-off constrains the optimization efficiency of critic-free algorithms.

To tackle this issue, Razin et al. (2024; 2025) reveals that raising the variance of reward signals can accelerate convergence, as higher reward variance ($\text{Var}(R)$) creates a steeper optimization land-

---

[*]Contributed equally.

[†]Work done while interning at Tencent.

[‡]Corresponding Author. `luckytyang@tencent.com, ding.wenbo@sz.tsinghua.edu.cn`

scape. However, for typically critic-free methods (e.g., GRPO and GSPO), advantages are computed by normalizing the selected subset, resulting in a fixed advantage variance of 1. Our theoretical analysis in Section 2.1 demonstrates that maximizing $\text{Var}(R)$ in critic-free methods imposes a *fixed* upper bound on the policy gradient norm, whereas maximizing advantage variance ($\text{Var}(A)$) introduces a *variable* upper bound positively correlated with the advantage variance. This indicates that maximizing $\text{Var}(A)$ has the potential to yield higher policy gradients within the core subset, thereby accelerating policy convergence toward the optimal trajectory.

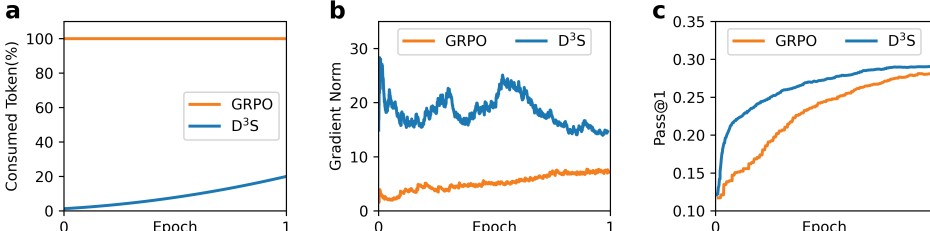

Figure 1: Comparison of training dynamics between $D^3S$ and the GRPO: (a) token consumption ratio, (b) gradient norms, and (c) Pass@1 scores. Compared to the original GRPO, the integration of $D^3S$ reduces token usage, accelerates policy convergence, and delivers superior performance.

Building on this insight, we propose **D**ynamic **D**ual-Level **D**own-**S**ampling (**D**$^3$**S**) framework, which operates on two levels. **First**, at the sample-level, instead of maximizing $\text{Var}(R)$, $D^3S$ selects samples by first estimating the group-relative advantage across the entire batch and then maximizing $\text{Var}(A)$ to identify the core subset for optimization. **Second**, at the token-level, we further consider both advantage and entropy metrics, proposing the product of advantage magnitude and entropy ($|A_{i,t}| \times H_{i,t}$) as a measure of token importance. The advantage magnitude reflects the relative significance of tokens, while entropy quantifies uncertainty, an essential factor for guiding exploratory reasoning in reasoning tasks (Wang et al., 2025). Moreover, to prevent the policy from overfitting to a limited set of high-signal data and compromising its generalization, we introduce a dynamic down-sampling schedule. Inspired by curriculum learning (Bengio et al., 2009), which progresses from simple to complex, the dynamic down-sampling schedule begins by prioritizing a smaller subset of high-signal samples and tokens to accelerate early-stage learning. As training progresses, the data scale gradually expands, incorporating more samples and tokens to improve generalization over the thorough data distribution beyond narrow high-signal subset.

To verify the effectiveness of $D^3S$, we conduct extensive experiments across various RL settings (i.e., GRPO and GSPO) on challenging mathematical reasoning tasks. The results demonstrate that incorporating $D^3S$ into GRPO and GSPO consistently improves performance while reducing sample and token requirements. A direct comparison of training dynamics, as illustrated in Figure 1, uses Qwen2.5-Math-7B (Yang et al., 2024) as the backbone and is trained on AIME24 (MAA, 2024). Compared to the original GRPO, $D^3S$ optimizes fewer than 20% of the tokens (Figure 1a) while achieving higher policy gradients (Figure 1b), leading to significantly faster convergence and superior Pass@1 scores on the test set (Figure 1c). Specifically, when using Qwen2.5-Math-7B as the backbone, GRPO with $D^3S$ achieves average improvements of 4.5 in Pass@1 and 3.7 in Pass@8 across seven datasets, compared to the original GRPO. Similarly, with Llama3.1-8B-Instruct (Grattafiori et al., 2024) as the backbone, GRPO with $D^3S$ outperforms the original by 3.3 in Pass@1 and 7.8 in Pass@8. Moreover, our analysis highlights the following key findings: (1) Both sample-level and token-level down-sampling effectively eliminate undifferentiated signals in the early training stages, accelerating policy convergence. (2) In the later training stages, the dynamic down-sampling schedule plays a crucial role in enhancing the generalization of $D^3S$. (3) $D^3S$ better manages entropy fluctuations, reflecting more stable policy training.

## 2 THEORETICAL ANALYSIS

### 2.1 PRELIMINARIES

Formally, let $x$ represent an input query from the dataset $\mathbb{D}$. A large language model with parameters $\boldsymbol{\theta}$ is defined as a policy $\pi_{\boldsymbol{\theta}}$. For each query, the policy $\pi_{\boldsymbol{\theta}}$ generates multiple responses, with a group

size of $G$. For the $i$-th response $y_i$, the number of tokens is denoted as $|y_i|$. GRPO (Shao et al., 2024) removes the need for a standalone value network by estimating group-relative advantages directly from $G$. The optimization objective is given as:

$$J_{\text{GRPO}}(\boldsymbol{\theta}) = \mathbb{E}_{x\sim\mathbb{D},\{y_i\}_{i=1}^G\sim\pi_{\boldsymbol{\theta}_{\text{old}}}(\cdot|x)} \tag{1}$$

$$\left[ \frac{1}{G}\sum_{i=1}^{G}\frac{1}{|y_i|}\sum_{t=1}^{|y_i|}\min(w_{i,t}(\boldsymbol{\theta})A_{i,t}, \text{clip}(w_{i,t}(\boldsymbol{\theta}), 1-\varepsilon, 1+\varepsilon)A_{i,t}) - \beta D_{\text{KL}}(\pi_{\boldsymbol{\theta}}\|\pi_{ref}) \right]$$

where the importance ratio is $w_{i,t}(\boldsymbol{\theta}) = \frac{\pi_{\boldsymbol{\theta}}(y_{i,t}|x,y_{i,<t})}{\pi_{\boldsymbol{\theta}_{\text{old}}}(y_{i,t}|x,y_{i,<t})}$ which will be clipped by hyper-parameter $\epsilon$. $\beta$ regulates the constraint on the KL-divergence $D_{\text{KL}}$. The group-normalized advantage is derived by standardizing the reward signal $R$ within each group:

$$A_i = \frac{R(x,y_i) - \frac{1}{G}\sum_{j=i}^{G}R(x,y_i)}{\text{std}(\{R(x,y_i)\}_{i=1}^{G})} \tag{2}$$

Building upon GRPO, Zheng et al. (2025) proposes GSPO, which utilizes the full sequence as context for next-token prediction. While adopting the same token-level group-relative advantage signal defined in Equation 2, GSPO further incorporates a novel importance ratio based on sequence likelihood. The optimization objective of GSPO is denoted as:

$$J_{\text{GSPO}}(\boldsymbol{\theta}) = \mathbb{E}_{x\sim\mathbb{D},\{y_i\}_{i=1}^G\sim\pi_{\boldsymbol{\theta}_{\text{old}}}(\cdot|x)} \tag{3}$$

$$\left[ \frac{1}{G}\sum_{i=1}^{G}\frac{1}{|y_i|}\sum_{t=1}^{|y_i|}\min(s_{i,t}(\boldsymbol{\theta})A_{i,t}, \text{clip}(s_{i,t}(\boldsymbol{\theta}), 1-\varepsilon, 1+\varepsilon)A_{i,t}) - \beta D_{\text{KL}}(\pi_{\boldsymbol{\theta}}\|\pi_{ref}) \right]$$

where the importance ratio is $s_{i,t}(\boldsymbol{\theta}) = \text{sg}\left[\left(\frac{\pi_{\boldsymbol{\theta}}(y_i|x)}{\pi_{\boldsymbol{\theta}_{\text{old}}}(y_i|x)}\right)^{\frac{1}{|y_i|}}\right] \cdot \frac{\pi_{\boldsymbol{\theta}}(y_{i,t}|x,y_{i,<t})}{\text{sg}[\pi_{\boldsymbol{\theta}_{\text{old}}}(y_{i,t}|x,y_{i,<t})]}$ with $\text{sg}[\cdot]$ denoting stopping gradient. The group-normalized advantage is same as Equation 2.

## 2.2 UPPER BOUNDS ON GRADIENT NORMS

We begin by analyzing the upper bound of the gradient norm in GRPO (also applicable to GSPO), as it determines the scale of policy updates and directly impacts training efficiency.

**Proposition 1.** *The gradient of $J_{GRPO}$ satisfies:*

$$\|\nabla_{\boldsymbol{\theta}}J_{GRPO}(\boldsymbol{\theta})\| \leq 4\gamma(x;\boldsymbol{\theta}) \tag{4}$$

*where $\gamma(x;\boldsymbol{\theta})$ denotes a static parameter related to input $x$ and model parameters $\boldsymbol{\theta}$.*

*Proof.* See Appendix B.1. ☐

Proposition 1 highlights a key property of group-based methods: the gradient norm is capped by a fixed upper bound, regardless of the explicit reward variance $\text{Var}(R)$. We then extend Proposition 1 to scenarios where optimization is performed on a subset sampled from the group $G$.

**Proposition 2.** *The gradient of $\hat{J}_{GRPO}$ which selects a subset from original rollout group with size $G$ satisfies:*

$$\|\nabla_{\boldsymbol{\theta}}\hat{J}_{GRPO}(\boldsymbol{\theta})\| \leq 3 \cdot 2^{\frac{1}{3}} \cdot \gamma(x;\boldsymbol{\theta}) \cdot (\sqrt{G-1})^{1/3} \cdot (Var(A'))^{1/3} \tag{5}$$

*where $Var(A')$ is the advantage variance of the selected subset from rollouts.*

*Proof.* See Appendix B.2. ☐

**Lemma 1.** *Let $A$ be a set of $M$ elements that is standardized as $\mathbb{E}[A] = 0, \text{Var}(A) = 1$. For any integer $N$ such that $2 \leq N \leq M$, there exists a subset $A' \subseteq A$ with $|A'| = N$ satisfying $\text{Var}(A') \geq 1$.*

*Proof.* See Appendix B.3. ☐

From Proposition 2, we can observe that under the down-sampling perspective, the gradient norm of GRPO has a *variable* upper bound, which is positively correlated with the advantage variance $\text{Var}(A')$ of the subset. Consequently, the strategy of maximizing $\text{Var}(R)$ (Razin et al., 2024; 2025; Xu et al., 2025) in group-based methods encounters two major limitations. **First**, even if $\text{Var}(R)$ is maximized within the selected subset, the advantages are still computed under the constraint of

a fixed $\text{Var}(A') = 1$ and cannot change the gradient norm upper bound. **Second**, the advantages are estimated within a smaller, biased subset of the original group, leading to unstable advantage estimation. This naturally motivates an alternative approach: compute normalized advantages using all samples from the original group, and then select a subset that maximizes the variance of these normalized advantages. Lemma 1 proves that a subset with a variance of at least 1 can be extracted from a set of normalized advantages. This indicates that such a subset leads to a higher upper bound on the gradient norm. The trends in Figure 1b provide empirical evidence supporting this property.

## 3 METHOD

In this study, we introduce the **Dynamic Dual-level Down-Sampling ($\text{D}^3\text{S}$)** framework, which improves training efficiency through a two-tier approach: a sample-level down-sampling strategy and a token-level selection mechanism.

### 3.1 SAMPLE-LEVEL: CROSS-GROUP ADVANTAGE-BASED DOWN-SAMPLING

Building on the insights from Section 2.2, rather than selecting subsets of rollouts based on maximizing $\text{Var}(R)$ (Razin et al., 2024; 2025; Xu et al., 2025), we propose a refined method that identifies a core subset of samples based on their group-relative estimated advantages. This approach prioritizes maximizing the variance of advantage signals within the selected subset. Formally, given a query $x$ and its rollouts $\mathcal{S}_{\text{query}} = \{(x, y_i, A_i) : i \in [1, G]\}$, the selected subset $\hat{\mathcal{S}}_{\text{query}}$ is defined as follows:

$$\hat{\mathcal{S}}_{\text{query}} = \underset{\hat{S} \subset \mathcal{S}_{\text{query}}, |\hat{S}| = N_{\hat{s}}}{\arg\max} \text{Var}(A_{\hat{S}}) \tag{6}$$

where $N_{\hat{s}}$ is the number of selected samples within the group $\mathcal{S}_{\text{query}}$, and $A_{\hat{S}} = \{A_1, A_2, ..., A_{|\hat{S}|}\}$ denotes the advantage set of $\hat{\mathcal{S}}_{\text{query}}$. Each advantage $A_i$ is defined as the mean of the token-level advantages $A_{i,t}$ over the sequence. Xu et al. (2025) shows that the subset maximizing variance can be efficiently selected from the $\binom{N_{\hat{S}}}{G}$ possible combinations, where $N_{\hat{S}} = N_{\hat{S},\text{pos}} + N_{\hat{S},\text{neg}}$. In our scenario, we adopt this implementation, where $N_{\hat{S},\text{pos}}$ refers to the samples with the highest positive advantages, while $N_{\hat{S},\text{neg}}$ corresponds to those with the lowest negative advantages.

Moreover, considering that gradient updates are performed in batches and there are significant advantage disparities across different groups (e.g., groups with all correct or incorrect predictions may result in uninformative zero advantages), we introduce a cross-group operation to select a high-variance subset from the entire batch. Formally, let $\mathcal{S}_{\text{batch}} = \{\mathcal{S}_{\text{query},b} : b \in [1, B]\}$ be a batch of rollouts, where $B$ is the batch size and $N = B \times N_{\hat{s}}$ is the total number of selected samples. The high-variance subset can then be obtained as:

$$\hat{\mathcal{S}}_{\text{batch}} = \underset{\hat{S} \subset \mathcal{S}_{\text{batch}}, |\hat{S}| = N}{\arg\max} \text{Var}(A_{\hat{S}}) \tag{7}$$

The cross-group operation retains the original distributional properties, as the advantages are pre-normalized within each group, and no additional normalization is applied.

### 3.2 TOKEN-LEVEL: ENTROPY-ADVANTAGE WEIGHTED SELECTION

Intuitively, responses often consist of a combination of easy tokens (where the model is both confident and accurate), neutral tokens (with minimal impact on outcomes), and critical tokens (where the model is uncertain and decisions significantly influence rewards). Policy entropy serves as a measure of the model's uncertainty (Cui et al., 2025) and as an indicator of potential performance gains. Wang et al. (2025) demonstrates that the top 20% of tokens with the highest entropy dominate the policy gradient. Treating all tokens equally during updates dilutes the gradient signal, akin to averaging out meaningful information with noise.

To this end, we propose a token-level selection mechanism that integrates generation entropy and its advantage into a unified importance metric for ranking tokens across all selected samples. Specifi-

cally, high-importance tokens are identified as follows:

$$H_{i,t} = -\sum_{j=1}^{V} \pi_\theta(\text{token}_j \mid x_i, y_{i,<t}) \log \pi_\theta(\text{token}_j \mid x_i, y_{i,<t}) \tag{8}$$

$$\mathcal{T} = \text{top}_{K\%}(y_{i,t}, y_{i,t} \in \hat{\mathcal{S}}, \text{key} = |A_{i,t}| \times H_{i,t})$$

where $H_{i,t}$ denotes the entropy of $t$-th token in $i$-th response, and $K$ indicates the proportion of selected tokens. High entropy $H_{i,t}$ indicates higher uncertainty in the model's decision at that token position, fostering exploration and enhancing policy diversity. Advantage $A_{i,t}$ quantifies the impact of token on policy improvement, where a larger $|A_{i,t}|$ reflecting greater potential—positive or negative—for optimization. During each update, only the top $K\%$ of tokens contribute to the gradient of $\theta$. By selecting tokens with high entropy and advantage, computation is focused on the most informative decision points, encouraging the policy to resolve uncertainty in reward-critical regions.

### 3.3 DYNAMIC DOWN-SAMPLING SCHEDULE

The sample-level and token-level strategies discussed in Sections 3.1 and 3.2 refine policy updates by prioritizing high-signal samples and tokens. Although this approach accelerates convergence by leveraging stronger gradients, it also heightens the risk of reward hacking or overfitting. The model might over-exploit a limited set of trajectories that seem highly informative in the early stages, but struggle to generalize as optimization progresses. To address this, we introduce a dynamic down-sampling schedule that progressively reduces the intensity of down-sampling as training advances.

Specifically, we employ a linear schedule to interpolate between the initial aggressive configuration $(N_{\text{init}}, K_{\text{init}})$ and the final milder configuration $(N_{\text{final}}, K_{\text{final}})$, based on the training progress $p \in [0, 1]$. We use $N_s^{(p)}$ to regulate the number of retained samples per query, while $K^{(p)}$ determines the proportion of retained tokens within each sample:

$$\left[N_s^{(p)}, K^{(p)}\right] = (1 - p) \cdot [N_{\text{init}}, K_{\text{init}}] + p \cdot [N_{\text{final}}, K_{\text{final}}] \tag{9}$$

At the beginning of training ($p = 0$), the model prioritizes the most informative rollouts and tokens to accelerate learning. As training advances ($p \to 1$), its focus broadens, incorporating more diverse signals to enrich learning and expand its scope. As empirical results show in Figure 3, variance-based selection enhances early performance but loses effectiveness over time and risks overfitting. In contrast, the dynamic schedule sustains both a fast convergence rate and consistent improvements. Our dynamic scheduling mechanism draws inspiration from curriculum learning. This dynamic curriculum ensures that the model can "generalize" to the entire data distribution while utilizing high-signal data, thus maintaining continuous performance improvement while avoiding overfitting.

### 3.4 D³S OPTIMIZATION OBJECTIVE

By integrating the sampling strategies outlined above, the objective function of D³S is expressed in Equation 10.

$$J_{\text{D}^3\text{S}}(\boldsymbol{\theta}) = \mathbb{E}_{x \sim \mathbb{D}, \{y_i\}_{i=1}^{G} \sim \pi_{\boldsymbol{\theta}_{\text{old}}}(\cdot|x)}$$

$$\frac{1}{|\hat{\mathcal{S}}|} \frac{1}{|\mathcal{T}|} \sum_{i=1}^{i \in \hat{\mathcal{S}}} \sum_{t=1}^{t \in \mathcal{T}} \left\{ \min\left[ w_{\boldsymbol{\theta}, i, t} A_{i,t}, \text{clip}(w_{\boldsymbol{\theta}, i, t}, 1 - \varepsilon, 1 + \varepsilon) A_{i,t} \right] - \beta D_{\text{KL}}(\pi_{\boldsymbol{\theta}} \| \pi_{ref}) \right\} \tag{10}$$

## 4 EXPERIMENT

### 4.1 CONFIGURATION

**Datasets and Evaluation**  We train each model using the DeepScaleR (Luo et al., 2025) dataset, which includes AIME problems from 1984 to 2023, AMC problems prior to 2023, and questions from other sources . During training, the AIME24 (MAA, 2024) is used as a validation set to monitor the out-of-domain performance of policy in real time. Evaluation is conducted on a diverse set of benchmarks, including AIME25 (MAA, 2025), AIME24 (MAA, 2024), AMC23 (MAA, 2023), GSM8K (Cobbe et al., 2021), MATH (Hendrycks et al., 2021), MinervaMath (Lewkowycz et al.,

2022) and OlympiadBench (He et al., 2024). For each question in these benchmarks, we generate 32 parallel outputs and compute Pass@k metrics. Rewards are assigned to the entire sequence based on the correctness of the answer, verified using math_verify tool (Kydlíček, 2025).

**Models**  We employ various models to systematically evaluate the proposed $D^3S$ framework. Qwen2.5-Math-7B (Yang et al., 2024), a pre-aligned model, is specifically optimized for mathematical tasks. Llama3.1-8B-Instruct (Grattafiori et al., 2024) serves as a general-purpose baseline model, while OpenMath2-Llama3.1-8B (Toshniwal et al., 2024), a fine-tuned variant of Llama3.1-8B-Instruct, is included for comparison. Besides, a smaller model, Qwen2.5-Math-1.5B(Yang et al., 2024), is utilized to assess adaptability across varying model scales. Each model is configured using its officially recommended settings as shown in Table 5.

**Baselines**  To assess the effectiveness of our approach, we integrate $D^3S$ into two popular RL algorithms: GRPO (Shao et al., 2024) and its variant, GSPO (Zheng et al., 2025), which enhances GRPO by improving sequence-level advantage estimation. Additionally, we compare our method with PODS (Xu et al., 2025), a down-sampling strategy via maximizing reward variance. To further analyze the impact of different stages of $D^3S$, we compare several variants: (1) $D^1S$, which applies sample-level down-sampling by maximizing advantage variance as defined in Equation 6; (2) $D^1S$ w/**Cross**, which incorporates cross-group operation as described in Equation 7; and (3) $D^2S$, which combines sample-level and token-level down-sampling but excludes the use of the dynamic down-sampling schedule, as defined in Equation 8. Parameters are listed in Table 6.

## 4.2 MAIN RESULTS

Table 1: Experimental results on various mathematical reasoning benchmarks using different model backbones. We report Pass@1/Pass@8 scores, computed from 32 parallel runs. The best results are highlighted in **bold**, while the second-best are underlined.

| Model | AIME24 | AIME25 | AMC23 | GSM8k | MATH | Minerva | Olympiad | Average |
|---|---|---|---|---|---|---|---|---|
| Qwen2.5-Math-7B | | | | | | | | |
| Base | 8.9/33.2 | 2.3/13.4 | 22.8/70.4 | 30.1/83.2 | 27.9/64.6 | 8.4/33.7 | 4.1/14.6 | 14.9/44.7 |
| GRPO | 13.2/37.6 | 5.5/21.6 | 47.0/83.5 | 64.9/94.3 | 48.5/70.2 | 19.8/45.0 | 9.7/19.8 | 29.8/53.1 |
| +PODS | 16.1/40.5 | 7.8/24.5 | 52.8/81.5 | **73.3**/95.0 | **53.0**/71.1 | 24.6/47.5 | **11.0**/20.7 | 34.1/54.4 |
| **+$D^3S$** | **20.3/48.2** | **7.9/25.8** | **54.4/87.1** | 71.3/**95.7** | 52.2/**71.5** | **25.0/48.2** | 10.7/20.8 | **34.3/56.8** |
| GSPO | 15.8/42.4 | 6.7/25.3 | 50.8/81.2 | 72.0/95.2 | 52.1/71.0 | 24.2/47.3 | 10.8/20.7 | 33.2/54.7 |
| +PODS | 15.4/40.9 | 6.5/22.9 | 51.9/81.6 | 72.9/95.3 | 52.9/71.1 | **25.0**/47.6 | 10.9/20.9 | 33.6/54.3 |
| **+$D^3S$** | **18.3/43.3** | **8.3/26.9** | **53.2/83.8** | **76.0/96.1** | **54.9/71.4** | 28.4/**51.1** | **11.5/21.1** | **35.8/56.2** |
| Qwen2.5-Math-1.5B | | | | | | | | |
| Base | 4.7/23.7 | 2.1/13.2 | 21.3/60.8 | 23.7/75.8 | 18.6/56.3 | 7.5/30.0 | 5.0/15.8 | 11.8/39.4 |
| GRPO | 10.0/28.3 | 6.1/19.9 | 46.2/77.0 | 77.3/**94.4** | 53.1/69.4 | 20.8/43.7 | 10.2/**18.9** | 32.0/50.2 |
| +PODS | **12.2**/30.6 | 5.9/22.3 | 47.4/75.1 | 77.0/94.2 | 53.2/69.3 | 21.8/**44.0** | 10.3/18.3 | 32.5/50.5 |
| **+$D^3S$** | 11.2/**32.2** | **6.9/24.0** | **48.6/79.7** | **77.5**/94.1 | **53.7/69.5** | **23.5/44.5** | **10.6**/18.4 | **33.1/51.8** |
| GSPO | 11.1/29.9 | 6.9/23.0 | **49.7/79.8** | 78.1/94.3 | 53.5/**69.4** | **23.0/44.1** | 10.5/**19.4** | 33.3/51.4 |
| +PODS | **12.3/32.9** | **6.9**/**24.0** | 47.8/77.1 | 77.4/94.1 | 53.4/69.4 | 22.5/43.3 | 10.2/18.7 | 32.9/51.4 |
| **+$D^3S$** | 11.4/32.8 | **8.2/25.2** | 48.4/79.1 | 78.0/94.1 | **54.0**/69.6 | 22.9/43.2 | **10.5**/19.0 | **33.3/51.9** |
| Llama3.1-8B-Instruct | | | | | | | | |
| Base | 1.7/10.9 | **0.4/2.8** | 15.0/47.3 | 57.7/92.8 | 29.3/55.9 | 14.7/38.5 | 2.2/8.2 | 17.3/36.6 |
| GRPO | 2.0/5.0 | 0.0/0.0 | 13.7/33.4 | 78.6/93.5 | 31.5/52.0 | 15.9/35.6 | 2.1/7.2 | 20.5/32.4 |
| +PODS | 2.8/9.9 | 0.3/2.5 | 14.5/38.1 | 77.1/93.3 | 31.5/52.6 | 16.3/38.0 | 2.2/7.6 | 20.7/34.6 |
| **+$D^3S$** | **5.3/20.7** | 0.1/0.8 | **20.3/50.8** | **79.0/95.0** | **35.9/59.2** | **22.5/44.3** | **3.3/10.7** | **23.8/40.2** |

Table 1 presents the alignment results of the different LLMs across seven reasoning benchmarks, using GRPO and GSPO as the base algorithms. Our observations are fourfold. **First**, the introduction of $D^3S$ consistently improves performance across all backbone models, demonstrating its strong adaptability across various types of backbones. Notably, $D^3S$ achieves the highest average scores of 35.8% for pass@1 and 56.8% for pass@8 on the Qwen2.5-Math-7B model. **Second**, $D^3S$ demonstrates a significant performance advantage over both the original method and PODS. For example, when Qwen2.5-Math-7B is used as the backbone, GRPO+$D^3S$ achieves an average improvement of 4.5 points in pass@1 and 3.7 points in pass@8 compared to the original GRPO. Similarly, with Llama3.1-8B-Instruct as the backbone, GRPO with $D^3S$ surpasses the original GRPO and

GRPO+PODS by 3.3 and 3.1 points on pass@1, and by 7.8 and 5.6 points on pass@8, respectively. **Third**, even for the strong baseline GSPO, incorporating D$^3$S still yields improvements of 2.6 and 1.5 points on pass@1 and pass@8, respectively. This demonstrates that the D$^3$S down-sampling strategy can be seamlessly generalized to and further enhance other algorithms leveraging group-relative advantages. **Fourth**, for the smaller-scale Qwen2.5-Math-1.5B model, D$^3$S still achieves the best average performance among all compared methods, showing its scalability and effectiveness across LLMs of varying sizes. We provide more experimental results, including pass@16 metrics, in Appendix D.3, which further validate the effectiveness of our approach.

## 4.3 Ablation Study of D$^3$S

D$^3$S integrates sample-level, token-level, and a dynamic down-sampling schedule to effectively train the policy model. To systematically evaluate the impact of each component in the D$^3$S strategy, we conduct a series of ablation studies by progressively incorporating more sophisticated selection strategies into the GRPO baseline, as listed in Table 2. Here, we utilize Qwen2.5-Math-7B as the base model. The notation for each method is explained in detail in Section 4.1.

Table 2: Ablation studies of different components in D$^3$S strategy. Performance is evaluated using Pass@8 across various benchmarks. The experiments utilize Qwen2.5-Math-7B as the base model and GRPO as the foundational algorithm.

| Model | AIME24 | AIME25 | AMC23 | GSM8k | MATH | Minerva | Olympiad | Average |
|---|---|---|---|---|---|---|---|---|
| Base | 8.9/33.2 | 2.3/13.4 | 22.8/70.4 | 30.1/83.2 | 27.9/64.6 | 8.4/33.7 | 4.1/14.6 | 14.9/44.7 |
| GRPO | 13.2/37.6 | 5.5/21.6 | 47.0/83.5 | 64.9/94.3 | 48.5/70.2 | 19.8/45.0 | 9.7/19.8 | 29.8/53.1 |
| +D$^1$S | 13.2/42.9 | 5.9/20.2 | 50.6/84.4 | 68.5/94.9 | 50.1/70.5 | 20.8/46.4 | 10.3/20.4 | 31.3/54.2 |
| +D$^1$S-C | 17.3/40.0 | 7.7/25.6 | 51.9/83.3 | **73.4**/95.4 | 52.8/70.9 | **25.0**/47.1 | 10.6/20.6 | 34.1/54.7 |
| +D$^2$S | 16.9/42.2 | 6.0/21.2 | 49.6/82.8 | 66.3/94.9 | 49.5/70.7 | 20.9/46.7 | 10.1/20.3 | 31.3/54.1 |
| **+D$^3$S** | **20.3/48.2** | **7.9/25.8** | **54.4/87.1** | 71.3/**95.7** | **52.2/71.5** | 23.4/**48.2** | **10.7/20.8** | **34.3/56.8** |

The ablation results reveal that the contribution of individual components is not strictly monotonic. For instance, D$^2$S occasionally underperforms D$^1$S on AIME24 but achieves greater improvements on AIME25. Nevertheless, the complete D$^3$S configuration consistently delivers the best performance. This provides strong evidence that the dual-level design, combined with a dynamic down-sampling schedule, effectively balances exploitation and exploration, leading to robust improvements across tasks. Beyond the Qwen2.5-Math-7B model, additional ablation studies and analyses on Llama3.1 models, detailed in Section D.1, further demonstrate the generalizability of the D$^3$S.

## 4.4 Ablation Study on Dynamic Scheduling Hyper-Parameter

Table 3: Ablation of parameter $N$ (number of selected samples) and $K$ (ratio of reserved tokens) used in dynamic scheduling (Equation 9) on GRPO algorithm in finetuning Qwen2.5-Math-7B. Results are presented in form of Pass@1/Pass@8 scores. The best results are highlighted in **bold** while the second-best are underlined.

| Configuration | AIME24 | AIME25 | AMC23 | GSM8k | MATH | Minerva | Olympiad | Average |
|---|---|---|---|---|---|---|---|---|
| D$^3$S | 20.3/48.2 | 7.9/25.8 | 54.4/**87.1** | 71.3/95.7 | 52.2/71.5 | **25.0**/48.2 | 10.7/20.8 | 34.3/**56.8** |
| $N = 8 \rightarrow 16$ | 20.1/46.2 | 7.9/23.2 | 56.5/85.0 | 76.1/**95.9** | 55.5/**71.7** | 23.0/46.9 | 11.1/21.0 | 35.8/55.7 |
| $N = 16 \rightarrow 32$ | 18.4/46.7 | 7.6/25.2 | 55.0/85.5 | 73.6/95.8 | 53.1/71.4 | 22.5/47.6 | 10.7/20.9 | 34.4/56.2 |
| $K = 5\% \rightarrow 30\%$ | 19.7/43.4 | **8.9**/25.6 | 57.4/81.3 | 77.6/95.7 | 55.9/71.7 | 23.6/47.6 | 11.3/21.2 | 36.3/55.2 |
| $K = 5\% \rightarrow 40\%$ | 18.1/41.1 | 7.6/24.6 | 55.2/82.5 | 72.5/95.4 | 54.1/71.5 | 21.7/46.9 | 10.2/20.8 | 34.2/54.7 |
| $K = 5\% \rightarrow 50\%$ | 17.5/44.5 | 7.6/25.8 | 52.0/82.8 | 74.4/95.6 | 54.2/**71.7** | 24.1/47.9 | 11.3/21.4 | 34.4/55.7 |
| $K = 10\% \rightarrow 20\%$ | 16.8/42.3 | 7.5/23.8 | 56.0/83.6 | 74.3/95.6 | 54.6/71.6 | 21.7/46.4 | 10.5/20.5 | 34.5/54.8 |
| $K = 10\% \rightarrow 30\%$ | 17.3/46.1 | 8.1/24.8 | 53.7/86.4 | 72.7/95.2 | 53.7/71.5 | 22.1/47.0 | 10.3/20.8 | 34.0/56.0 |
| $K = 10\% \rightarrow 40\%$ | 18.6/45.0 | 7.7/26.1 | 52.5/83.2 | 70.2/95.6 | 52.2/71.4 | 22.5/48.2 | 10.4/20.6 | 33.4/55.7 |
| $K = 10\% \rightarrow 50\%$ | 17.9/44.5 | 8.0/**27.8** | 52.3/83.6 | 72.7/95.6 | 53.9/71.5 | 22.2/47.2 | 10.7/20.7 | 34.0/55.8 |

To assess the sensitivity of D$^3$S dynamic down-sampling schedule to its key hyper-parameters ($N_{init}, N_{final}, K_{init}, K_{final}$), we conduct a detailed ablation study. We use Qwen2.5-Math-7B as the backbone, and the results are presented in Table 3. Our baseline D$^3$S configuration (used for main results in Section 4.2) sets $G = 32$ (large enough in favor of GRPO advantage estimation, fixed in this ablation), $N = 8 \rightarrow 32$ and $K = 5\% \rightarrow 20\%$, achieving an average Pass@1/Pass@8 of 34.84/56.76. This configuration heuristically draws inspiration from empirical conclusion from

Wang et al. (2025). The ablation results show that $D^3S$ is not overly sensitive to these choices. When testing different sample-level configurations, such as $N = 8 \rightarrow 16$ (35.74/55.70) and $N = 16 \rightarrow 32$ (34.41/56.16), the average performance remains remarkably stable and comparable to our baseline. We also explored a wide range of token ratios. All configurations maintain strong performance, while $K = 5\% \rightarrow 30\%$ achieves even higher Pass@1 on multiple benchmarks, which indicates training might benefit from more "low-signal" data. This improvement further shows the advantage of "diversity-increasing" schedule strategy of $D^3S$.

This analysis demonstrates that the effectiveness of $D^3S$ is robust across a reasonable range of hyperparameters. $D^3S$ does not require extensive, task-specific tuning to achieve significant performance and efficiency gains.

## 4.5 TRAINING EFFICIENCY

Table 4: Comparison of training efficiency. We assess the performance gains brought by integrating $D^3S$ into the GRPO and GSPO, along with the time acceleration needed to achieve these gains.

| Methods | $D^3S$ vs GRPO | | $D^3S$ vs GSPO | |
|---|---|---|---|---|
| | Avg@32 | Time | Avg@32 | Time |
| Qwen2.5-Math-7B | **+6%** | **2.04×** | **+17%** | **5.51×** |
| Qwen2.5-Math-1.5B | +4% | 1.57× | +2% | 1.10× |

Table 4 highlights the remarkable training efficiency of $D^3S$. For instance, on the Qwen2.5-Math-7B model, $D^3S$ achieves an average accuracy (Avg@32) that is 6% higher than GRPO, while reducing the training time by half to reach the same performance level (2.04× speedup). The benefits are even more pronounced compared to GSPO, with a 5.51× faster training speed and a 17% accuracy improvement. On the smaller Qwen2.5-Math-1.5B model, $D^3S$ delivers a 4% accuracy boost alongside a 1.57× speedup. These findings underscore $D^3S$'s ability to not only accelerate model convergence but also enhance final performance, particularly for larger models.

## 4.6 DYNAMIC DOWN-SAMPLING SCHEDULE MITIGATES OVERFITTING

To better understand how $D^3S$ impacts policy optimization, we track key metrics during the training process. The first metric, sample usefulness rate (SUR), measures the proportion of groups in each batch with non-zero advantages, reflecting the percentage of samples that actively contribute to policy updates. The second metric, *KL divergence* ($D_{KL}$), quantifies the difference between the policy and reference model distributions. A higher $D_{KL}$ indicates greater divergence from the reference model, which may signal an increased risk of overfitting.

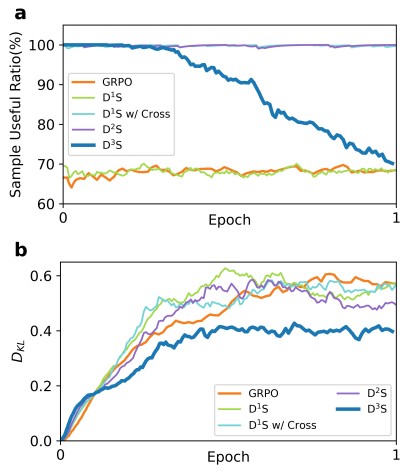

Figure 2 illustrates the training dynamics under various $D^3S$ settings, as detailed in Section 4.1. Four key observations emerge: **First**, as shown in Figure 2a, both the original GRPO and $D^1S$ (without cross-group operation) maintain a SUR of approximately 70%, with minor fluctuations. **Second**, introducing cross-group operation (e.g., $D^1S$ w/Cross and $D^2S$) significantly boosts the SUR to nearly 100%, indicating that cross-group mechanism

Figure 2: Training dynamics of (a) sample usefulness and (b) KL divergence.

effectively filter out ambiguous data within the training batch. **Third**, as shown in Figure 2b, the $D_{KL}$ curves for $D^1S$, $D^1S$ w/Cross, and $D^2S$ rise more steeply in the early stages compared to GRPO but eventually converge to similar values. This suggests that while these methods initially deviate more from the reference model, they ultimately reach comparable limits, potentially increasing the risk of overfitting. **Fourth**, the SUR gradually declines from 100% to 70% as training progresses, aligning with the design goal of dynamic down-sampling—incrementally increasing samples and tokens to mitigate overfitting in later stages. The $D_{KL}$ curve also demonstrates that $D^3S$ achieves smaller deviations from the reference model compared to other methods.

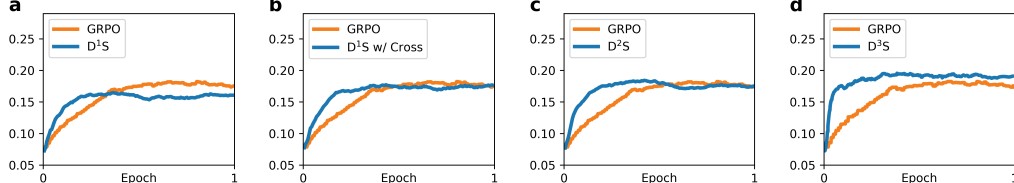

Figure 3: The Avg@32 test performance of AIME24 on Qwen2.5-Math-7B under various settings. Methods without a dynamic down-sampling schedule (a,b,c) accelerate convergence in the early stages but suffer from overfitting later. In contrast, the dynamic down-sampling schedule (d) not only accelerates convergence initially but also outperforms other methods in the later stages.

Figure 3 provides a further comparison of Avg@32 performance under different down-sampling configurations. It can be observed that down-sampling strategies without a dynamic schedule consistently accelerate convergence but exhibit varying degrees of overfitting, with their Avg@32 accuracies eventually being surpassed by GRPO in the later stages of training (Figure 3a, b, c). In contrast, $D^3S$ (Figure 3d), which incorporates a dynamic schedule, not only accelerates convergence in the early stages but also achieves significantly better results in the later stages. This highlights the critical role of dynamic down-sampling schedule in mitigating overfitting.

## 4.7 ENTROPY ANALYSIS OF $D^3S$

We investigate entropy dynamics across various base models and RL algorithms to understand learning behaviors. As illustrated in Figures 4a and 4b, $D^3S$ consistently achieves lower and more stable policy entropy compared to baseline algorithms. This improvement stems from the token-level selection mechanism described in Section 3.2. By prioritizing updates on tokens with a high advantage-entropy product ($|A_{i,t}| \times H_{i,t}$), $D^3S$ focuses learning on resolving high-impact uncertainties. This targeted strategy enables the model to converge more efficiently toward a decisive policy, as evidenced by its reduced average entropy.

Conversely, the Llama3.1-8B-Instruct model exhibits a distinct behavior, as illustrated in Figure 4c. We hypothesize that this phenomenon arises from the base capabilities of Llama3.1 and its lack of adaptation to mathematical reasoning tasks, which leads $D^3S$ to promote more effective and productive exploration. The baseline GRPO algorithm only begins to slowly increase entropy when training is nearly halfway complete, indicating inefficient and insufficient exploration. In contrast, $D^3S$, through its token-level selection mechanism (prioritizing tokens with high $|A_{i,t}| \times H_{i,t}$), is able to encourage productive exploration both earlier and more effectively, thereby enabling the model to escape local optima more efficiently. To validate this hypothesis, we introduce OpenMath2-Llama3.1-8B (Toshniwal et al., 2024), a model fine-tuned specifically for mathematical tasks, as a point of comparison. As illustrated in Figure 4d, $D^3S$ exhibits entropy dynamics consistent with those observed in Figures 4a and 4b, reinforcing our hypothesis. In Figure 4d, while the baseline GRPO still shows a sharp and unstable entropy spike early in training, $D^3S$ ensures a smoother and more stable learning trajectory with consistently low entropy. Comparison between results in Figure 4c and 4d strongly indicates that $D^3S$ is able to adaptively encourage exploration (earlier,

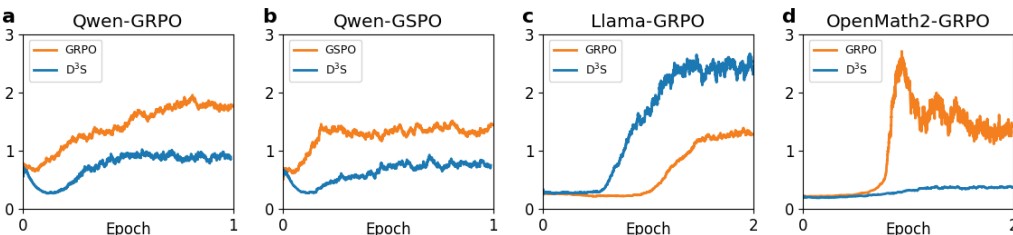

Figure 4: Entropy dynamics of different base models and RL algorithms. $D^3S$ effectively balances exploration and exploitation, fostering confident policies in well-aligned backbones (a, b, and d) while driving essential exploration in less-aligned ones (c).

sharper entropy increment) on less-aligned models and accelerate exploitation (lower entropy curve) on well-aligned models. Additional analyses are provided in Appendix D.1.

In conclusion, the entropy dynamics reveal that $D^3S$ effectively balances exploration and exploitation, fostering confident policies in aligned models while encouraging necessary exploration in less-aligned ones. This further underscores the robustness of the proposed framework.

## 5 RELATED WORKS

**Data Selection for Enhancing Training Efficiency**   As models and datasets scale, selecting data with high informational value becomes crucial for improving training efficiency. Razin et al. (2024; 2025) suggests that increasing reward signal variance sharpens the optimization landscape, thereby accelerating convergence. Lu et al. (2024) introduces SEAM, an automated metric for quantifying PM-RM differences, which enhances training reliability by filtering out samples where RM misjudges PM outputs. Several works focus on curriculum-based selection. Curry-DPO (Pattnaik et al., 2024) and LPPO (Chen et al., 2025) dynamically adjust sample weighting based on difficulty or learning progress. Similarly, Sun et al. (2025) proposes DOTS, which filters questions based on a heuristic difficulty metric (pass rate) estimated by an auxiliary predictor. Coresets (Mirzasoleiman et al., 2020) aim to select a weighted subset to provably approximate the full dataset's gradient. Unlike these approaches, $D^3S$ does not rely on heuristic difficulty or gradient approximation. Instead, our sample-level strategy maximizes the Advantage Variance of rollout groups, which directly maximizes the upper bound of the policy gradient norm for efficient optimization.

**Token-Level Entropy Utilization**   Entropy reflects the unequal importance of tokens within a sequence. Wang et al. (2025) observes that policy gradients are sparse and driven by a minority of high-entropy tokens. To address the risk of entropy collapse caused by high-signal tokens, Cui et al. (2025) identifies tokens with high covariance between probability and advantage as the cause of instability, proposing Clip-Cov to suppress their updates. Similarly, Hao et al. (2025) introduces STEER to stabilize training by down-weighting tokens that contribute significantly to entropy change. Wen et al. (2024) introduces ETPO to improve entropy-regularized credit assignment. Similarly, Shen (2025) proposes AEnt, which calculates clamped entropy over a subset of high-probability tokens and employs an adaptive coefficient to balance the entropy reward. While these methods focus on *stability* by suppressing high-impact tokens, $D^3S$ adopts a distinct philosophy of *efficiency* via prioritization. Prioritizing high-signal data has been utilized by Prioritized Experience Replay (PER) (Schaul et al., 2016), which samples and replays uncertain transitions basing on their TD-error magnitude. Cheng et al. (2025) shapes GRPO advantages with entropy to improve exploration, instead of using entropy regularization. In our study, $D^3S$ integrates entropy with advantage magnitude ($|A| \times H$) to precisely locate tokens that are both critical and uncertain. Rather than suppressing these high-signal tokens, $D^3S$ explicitly prioritize them to accelerate convergence, together with employing a dynamic schedule to mitigate the overfitting risks.

## 6 CONCLUSION

In this study, we reveal that the theoretical upper bound of the policy gradient norm in group-relative advantage-based algorithms (e.g., GRPO) is positively correlated with advantage variance. Leveraging this insight, we introduce the Dynamic Dual-level Down-Sampling ($D^3S$) framework to enhance training efficiency. At the sample-level, $D^3S$ selects rollout responses to maximize advantage variance, while at the token-level, it prioritizes tokens with a high product of entropy and advantage magnitude, directing updates to regions where the model is both uncertain and impactful. To mitigate overfitting, $D^3S$ adopts a dynamic down-sampling schedule inspired by curriculum learning, gradually relaxing sampling criteria over time. Extensive experiments on the Qwen2.5 and Llama3.1 models show that $D^3S$ consistently surpasses baseline methods across diverse reasoning benchmarks. Ablation studies and training dynamic analyses reveal that the synergy between $D^3S$ components is crucial and $D^3S$ is robust against introduced hyper-parameters. These results highlight the significance of fine-grained sample utilization in RLHF and emphasize the pivotal role of entropy in managing the exploration-exploitation trade-off.

ETHICS STATEMENT

Our study introduces the Dynamic Dual-Level Down-Sampling ($D^3S$) framework to improve the efficiency and performance of RL algorithms. We affirm that this research raises no significant ethical concerns. All methodologies strictly adhere to ethical standards and responsible research practices. The datasets used are publicly available, widely recognized within the research community, and utilized in full compliance with their terms and conditions. Additionally, we declare no conflicts of interest, sponsorships, or external influences that could compromise the integrity of this work. In line with our commitment to transparency and reproducibility, we will release our code publicly to facilitate further research and innovation in this domain.

REPRODUCIBILITY STATEMENT

Experimental details are presented in Section 4.1. The code and data used in this study are included in the supplementary materials and will be publicly released. Proofs for the main theoretical results (Propositions 1, 2, and Lemma 1) can be found in Appendices B.1, B.2, and B.3, respectively.

ACKNOWLEDGMENT

This work was supported by the WeChat Search team, Tencent, which provided necessary research environment for this work. This work was motivated by challenges in AI search systems at WeChat. This work was also supported by Shenzhen Science and Technology Program (No. KJZD20240903100905008).

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

## A  USAGE OF LARGE LANGUAGE MODELS

In this study, we leverage LLMs to summarize and refine academic papers, ensuring clarity, precision, grammatical accuracy, and correct spelling. These models also offer suggestions to improve coherence and readability. Our aim is to elevate the efficiency and quality of academic writing.

## B  PROOF

### B.1  PROOF OF PROPOSITION 1

*Proof of Proposition 1.*  We analyze the gradient at the reference policy without considering the clipping operation, as it is inactive when the importance ratios equal 1.

The gradient becomes:

$$\nabla_{\boldsymbol{\theta}} J_{\text{GRPO}}(\boldsymbol{\theta}) = \mathbb{E}_{x \sim \mathbb{D}, y_i{}_{i=1}^G \sim \pi_{\boldsymbol{\theta}}(\cdot|x)} \left[ \frac{1}{G} \sum_{i=1}^{G} \frac{1}{|y_i|} \sum_{t=1}^{|y_i|} A_{i,t} \nabla_{\boldsymbol{\theta}} \log \pi_{\boldsymbol{\theta}}(y_{i,t}|x, y_{i,<t}) \right] \tag{11}$$

$$= \mathbb{E}_{x \sim \mathbb{D}, y_i{}_{i=1}^G \sim \pi_{\boldsymbol{\theta}}(\cdot|x)} \left[ \frac{1}{G} \sum_{i=1}^{G} \frac{1}{|y_i|} \sum_{t=1}^{|y_i|} A_{i,t} \nabla_{\boldsymbol{\theta}} \log \text{softmax}(z_{\boldsymbol{\theta}}(y_{i,t}|x, y_{i,<t})) \right] \tag{12}$$

$$= \mathbb{E}_{x \sim \mathbb{D}, y_i{}_{i=1}^G \sim \pi_{\boldsymbol{\theta}}(\cdot|x)}$$

$$\left[ \frac{1}{G} \sum_{i=1}^{G} \frac{1}{|y_i|} \sum_{t=1}^{|y_i|} A_{i,t} (I_{i,t}^{\text{one-hot}} - \pi_{\boldsymbol{\theta}}(\cdot|x, y_{i,<t}))^T \nabla_{\boldsymbol{\theta}} z_{\boldsymbol{\theta}}(y_{i,t}|x, y_{i,<t}) \right] \tag{13}$$

In following analysis, we simplify the average of token-wise advantages $\sum_{t=1}^{|y_i|} A_{i,t}$ into sequence-wise advantage $A_i$. On the other hand, it is also a description of the more commonly used sequence-wise outcome-rewarded GRPO implementations in math tasks. Formally:

$$\nabla_{\boldsymbol{\theta}} J_{\text{GRPO}}(\boldsymbol{\theta}) = \mathbb{E}_{x \sim \mathbb{D}, y_i{}_{i=1}^G \sim \pi_{\boldsymbol{\theta}}(\cdot|x)} \left[ \frac{1}{G} \sum_{i=1}^{G} A_i \nabla_{\boldsymbol{\theta}} \log \pi_{\boldsymbol{\theta}}(y_i|x) \right] \tag{14}$$

where $\nabla_{\boldsymbol{\theta}} \log \pi_{\boldsymbol{\theta}}(y_i|x) = \frac{1}{|y_i|} \sum_{t=1}^{|y_i|} \nabla_{\boldsymbol{\theta}} \log \pi_{\boldsymbol{\theta}}(y_{i,t}|x, y_{i,<t})$.

For any $c > 0$, we decompose the sum based on the magnitude of the standardized advantage:

$$\nabla_{\boldsymbol{\theta}} J_{\text{GRPO}}(\boldsymbol{\theta}) = \mathbb{E} \left[ \frac{1}{G} \sum_{i:|A_i| \leq c}^{G} A_i \nabla_{\boldsymbol{\theta}} \log \pi_{\boldsymbol{\theta}}(y_i|x) \right] \tag{I}$$

$$+ \mathbb{E} \left[ \frac{1}{G} \sum_{i:|A_i| > c}^{G} A_i \nabla_{\boldsymbol{\theta}} \log \pi_{\boldsymbol{\theta}}(y_i|x) \right] \tag{II}$$

Bounding term I: For samples with $|A_i| \leq c$:

$$\|(\text{I})\| \leq \mathbb{E} \left[ \frac{1}{G} \sum_{i:|A_i| \leq c}^{G} |A_i| \cdot \|\nabla_{\boldsymbol{\theta}} \log \pi_{\boldsymbol{\theta}}(y_i|x)\| \right] \tag{15}$$

$$\leq \mathbb{E} \left[ \frac{1}{G} \sum_{i:|A_i| \leq c}^{G} c \cdot \|(I_i^{\text{one-hot}} - \pi_{\boldsymbol{\theta}}(\cdot|x))^T \nabla_{\boldsymbol{\theta}} z_{\boldsymbol{\theta}}(y_i|x)\| \right] \tag{16}$$

$$\leq \mathbb{E} \left[ \frac{1}{G} \sum_{i:|A_i| \leq c}^{G} c \cdot \|(I_i^{\text{one-hot}} - \pi_{\boldsymbol{\theta}}(\cdot|x))^T\| \cdot \|\nabla_{\boldsymbol{\theta}} z_{\boldsymbol{\theta}}(y_i|x)\| \right] \tag{17}$$

Considering non-negative property of one-hot vector and likelihood vector, we can get:

$$\|I_i^{\text{one-hot}} - \pi_{\boldsymbol{\theta}}(\cdot|x)\| \leq \|I_i^{\text{one-hot}} - \pi_{\boldsymbol{\theta}}(\cdot|x)\|_1 \leq 2 \tag{18}$$

where $\| \cdot \|_1$ denotes the L1 norm.

Apply Equation 18 to Equation 17, we can get:

$$\|(\mathrm{I})\| \leq c \cdot 2 \cdot \gamma(x; \boldsymbol{\theta}) \tag{19}$$

where $\gamma(x; \boldsymbol{\theta})$ denotes a static parameter related to input query $x$ and model parameters $\boldsymbol{\theta}$.

Similarly, we apply Equation 18 and $\gamma(x; \boldsymbol{\theta})$ in bounding term II:

$$\|(\mathrm{II})\| = \mathbb{E}\left[\frac{1}{G} \sum_{i:|A_i|>c}^{G} A_i \nabla_{\boldsymbol{\theta}} \log \pi_{\boldsymbol{\theta}}(y_i|x)\right] \tag{20}$$

$$\leq \mathbb{E}\left[\frac{1}{G} \sum_{i:|A_i|>c} |A_i||\nabla_{\boldsymbol{\theta}} \log \pi_{\boldsymbol{\theta}}(y_i|x)|\right] \tag{21}$$

$$\leq 2 \cdot \gamma \cdot \mathbb{E}\left[\frac{1}{G} \sum_{i:|A_i|>c}^{G} |A_i|\right] \tag{22}$$

$$= 2 \cdot \gamma \cdot \mathbb{E}[|A_i| \cdot \mathbf{1}_{|A_i|>c}] \tag{23}$$

For samples with $|A_i| > c$, we use the refined Chebyshev inequality. Since $A_i$ is group-normalized with $\mathbb{E}[A_i] = 0$ and $\mathrm{Var}(A_i) = 1$, we have $\mathbb{E}[A_i^2] = 1$. By the refined Chebyshev bound:

$$\mathbb{E}[|A_i| \cdot \mathbf{1}_{|A_i|>c}] \leq \frac{\mathbb{E}[A_i^2]}{c} = \frac{1}{c} \tag{24}$$

Therefore:

$$\|(\mathrm{II})\| \leq 2 \cdot \gamma(x; \boldsymbol{\theta}) \cdot \frac{1}{c} \tag{25}$$

Combining the bounds:

$$\|\nabla_{\boldsymbol{\theta}} J_{\mathrm{GRPO}}(\boldsymbol{\theta})\| \leq \|(\mathrm{I})\| + \|(\mathrm{II})\| \tag{26}$$

$$\leq c \cdot 2 \cdot \gamma(x; \boldsymbol{\theta}) + \frac{2 \cdot \gamma(x; \boldsymbol{\theta})}{c} \tag{27}$$

$$= 2\gamma(x; \boldsymbol{\theta})\left(c + \frac{1}{c}\right) \tag{28}$$

The right-hand side objective is minimized when $c = 1$, we can obtain:

$$\|\nabla_{\boldsymbol{\theta}} J_{\mathrm{GRPO}}(\boldsymbol{\theta})\| \leq 2 \cdot \gamma(x; \boldsymbol{\theta}) \cdot 2 = 4\gamma(x; \boldsymbol{\theta}) \tag{29}$$

$\square$

## B.2 Proof of Proposition 2

*Proof of Proposition 2.* Considering Equation 24, the premise for it to be valid is $\mathbb{E}[A] = 0, \mathrm{Var}(A) = 1$ thus $\mathbb{E}[A^2] = \mathrm{Var}(A) + \mathbb{E}[A]^2 = 1$, which is property of standardized advantages. Thus, applying $\mathbb{E}[A^2] = 1$ and $|A_i| > c$, we can proof Equation 24 as:

$$\mathbb{E}[|A_i| \cdot \mathbf{1}_{|A_i|>c}] = \int_{|A_i|>c} |A_i|f(A)dA \tag{30}$$

$$\leq \int_{|A_i|>c} |A_i| \cdot \frac{|A_i|}{c} \cdot f(A)dA \tag{31}$$

$$= \frac{1}{c} \int_{|A_i|>c} |A_i|^2 f(A)dA \tag{32}$$

$$\leq \frac{1}{c} \int_{-\infty}^{\infty} |A_i|^2 f(A)dA \tag{33}$$

$$= \frac{\mathbb{E}[A_i^2]}{c} = \frac{1}{c} \tag{34}$$

However, considering $|A|$-driven down-sampling, $\mathbb{E}[A'^2] = 1$ no longer holds for subset $A'$.

We first estimate the upper bound of $|A_i|$. In GRPO, advantages are standardized as Equation 2. When reward signals are fixed in set $0, 1$, the maximum of $|A_i|$ within a group of size $G$ can be obtained when only 1 sample $i$ is rewarded with $1/0$ with other $G - 1$ samples rewarded with $0/1$ correspondingly. Formally:

$$|A_i|_{\max} = \frac{R_i - \mathbb{E}[R]}{\mathrm{std}[R]} = \frac{1 - \frac{1}{G}}{\sqrt{\mathbb{E}[R^2] - \mathbb{E}[R]^2}} \tag{35}$$

$$= \frac{1 - \frac{1}{G}}{\frac{\sqrt{G-1}}{G}} = \sqrt{G - 1} \tag{36}$$

Apply Equation 36 to Equation 23, we can get:

$$\mathbb{E}[|A_i'| \cdot \mathbf{1}_{|A_i'| > c}] \leq \sqrt{G - 1} \cdot \mathcal{P}(|A_i'| > c) \tag{37}$$

$$\|(\mathrm{II})\| \leq 2 \cdot \gamma(x; \boldsymbol{\theta}) \cdot \sqrt{G - 1} \cdot \frac{\mathrm{Var}[A']}{c^2} \tag{38}$$

$$\|\nabla_{\boldsymbol{\theta}} J_{\mathrm{GRPO}}(\boldsymbol{\theta})\| \leq 2 \cdot \gamma(x; \boldsymbol{\theta}) \left( c + \sqrt{G - 1} \cdot \frac{\mathrm{Var}(A')}{c^2} \right) \tag{39}$$

where we try to choose optimal $c$:

$$\frac{d}{dc} \left( c + \sqrt{G - 1} \cdot \frac{\mathrm{Var}(A')}{c^2} \right) = 1 - 2\sqrt{G - 1} \cdot \frac{\mathrm{Var}(A')}{c^3} = 0 \tag{40}$$

$$c^3 = 2\sqrt{G - 1} \cdot \mathrm{Var}(A') \tag{41}$$

$$c^* = \left[ 2\sqrt{G - 1} \cdot \mathrm{Var}(A') \right]^{1/3} \tag{42}$$

Thus

$$\|\nabla_{\boldsymbol{\theta}} J_{\mathrm{GRPO}}(\boldsymbol{\theta})\| \leq 3 \cdot 2^{\frac{1}{3}} \cdot \gamma(x; \boldsymbol{\theta}) \cdot (\sqrt{G - 1})^{1/3} \cdot (\mathrm{Var}(A'))^{1/3} \tag{43}$$

$\square$

### B.3 PROOF OF LEMMA 1

*Proof of Lemma 1.* We proceed by backward induction on $N$, starting from $N = M$ down to $N = 2$.

Base case ($N = M$): Take $A' = A$. Then $\mathrm{Var}(A') = 1 \geq 1$.

Inductive step: Assume for some $n + 1$ with $2 \leq n + 1 \leq M$ that there exists a subset $A'_{n+1} \subseteq A$ with $|A'_{n+1}| = n + 1$ and $\mathrm{Var}(A'_{n+1}) \geq 1$. We will show that there exists a subset $A'_n \subseteq A'_{n+1}$ with $|A'_n| = n$ and $\mathrm{Var}(A'_n) \geq 1$ by removing an element $a \in A'_{n+1}$ to form $A'_n = A'_{n+1} \setminus \{a\}$.

Considering definition of variance:
$$n \mathrm{Var}(A'_n) + (a - \mu_{n+1})^2 = (n + 1) \mathrm{Var}(A'_{n+1}) \tag{44}$$

We need $\mathrm{Var}(A'_n) \geq 1$, thus:
$$\frac{n + 1}{n} \mathrm{Var}(A'_{n+1}) - \frac{(a - \mu_{n+1})^2}{n} \geq 1 \tag{45}$$

Thus we need to find an $a \in A'_{n+1}$ that
$$(a - \mu_{n+1})^2 \leq (n + 1) \mathrm{Var}(A'_{n+1}) - n \tag{46}$$

When $\mathrm{Var}(A'_{n+1}) = 1$, it is easy to find an $a$ makes $(a - \mu_{n+1})^2 \leq (n + 1) - n = 1 = \mathrm{Var}(A'_{n+1})$ hold since $\mathrm{Var}(A'_{n+1})$ is the average of $(a - \mu_{n+1})^2$, the distance of elements to average, over $A'_{n+1}$.

When $\mathrm{Var}(A'_{n+1}) > 1$, we assume there is no $a$ makes Equation 46 hold. Formally:
$$\forall a \in A'_{n+1}, (a - \mu_{n+1})^2 > (n + 1) \mathrm{Var}(A'_{n+1}) - n \tag{47}$$

Thus

$$\text{Var}(A'_{n+1}) > (n+1)\text{Var}(A'_{n+1}) - n \tag{48}$$

$$1 > \text{Var}(A'_{n+1}) \tag{49}$$

which is conflict with $\text{Var}(A'_{n+1}) \geq 1$.

So assumption 47 does not hold, which means when $\text{Var}(A'_{n+1}) > 1$:

$$\exists a \in A'_{n+1}, (a - \mu_{n+1})^2 \leq (n+1)\text{Var}(A'_{n+1}) - n \tag{50}$$

So Equation 46 always holds when $\text{Var}(A'_{n+1}) \geq 1$ and $\text{Var}(A'_n) \geq 1$. By induction, Lemma 1 holds for all $N$ with $2 \leq N \leq M$.

$\square$

## C  TRAINING HYPER-PARAMETERS

Table 5 and 6 list hyper-parameters used in experiments.

Table 5: Generation configuration of different base models according to their official release.

| Model | Epoch | Temperature | Top-p | Learning rate |
|---|---|---|---|---|
| Qwen2.5-Math-7B | 1 | 1.0 | 0.9 | $5e^{-7}$ |
| Qwen2.5-Math-1.5B | 1 | 1.0 | 0.9 | $5e^{-7}$ |
| Llama3.1-8B-Instruct | 2 | 0.6 | 0.9 | $1e^{-8}$ |
| OpenMath2-Llama3.1-8B | 2 | 0.7 | 0.95 | $2e^{-7}$ |

Table 6: Hyper parameters used in finetuning.

| Parameter | Description | Value |
|---|---|---|
| $G$ | Group size of GRPO and GSPO | 32 |
| $\epsilon$ | Clip thereshold of importance ratio | 0.2 |
| $N_{\text{init}}$ | Init size of $D^3S$ selected responses | 8 |
| $N_{\text{final}}$ | Final size of $D^3S$ selected responses | 32 |
| $K_{\text{init}}$ | Init ratio of $D^3S$ selected tokens | 5% |
| $K_{\text{final}}$ | Final ratio of $D^3S$ selected tokens | 20% |

## D  DETAILED EXPERIMENT RESULTS

### D.1  ANALYSIS ABOUT EXPERIMENT RESULTS OF LLAMA MODEL

In our ablation studies, a noteworthy phenomenon emerged when applying the $D^3S$ framework to the general-purpose Llama3.1-8B-Instruct model, which has not been specifically aligned for mathematical reasoning. As detailed in Table 7, strategies based on intra-group sampling ($D^1S$ and $D^3S$-I) demonstrated markedly superior performance compared to their cross-group counterparts ($D^1S$-C and $D^3S$). Specifically, $D^3S$-I, which omits the cross-group sampling component, achieved an average Pass@1/Pass@8 score of 24.0/40.2, surpassing the full $D^3S$ configuration.

The training dynamics, depicted in Figure 5, provide a clear explanation for this discrepancy. The cross-group $D^3S$ strategy (dark blue line) induced extremely high-variance and unstable policy gradients, as shown in Figure 5a, accompanied by a sharp increase in $D_{\text{KL}}$ (Figure 5c). This suggests that for an unaligned model with highly heterogeneous output quality across different prompts, the global selection mechanism over-concentrates the learning signal on a few outlier samples, leading to an unstable optimization process. In contrast, $D^3S$-I (purple line) maintained a policy gradient that was both high in magnitude and remarkably stable, with a much milder policy distribution migration. This indicates that for unaligned models, providing a stable, localized learning signal within each prompt's context is more effective than pursuing the globally maximal signal.

This behavior changes when the framework is applied to OpenMath2-Llama3.1-8B, a domain-aligned version of the same base model. The results in Table 7 show that the performance gap

Table 7: Ablation study by incrementally applying different part of strategies of $D^3S$ to Llama3.1-8B-Instruct and OpenMath2-Llama3.1-8B. Performance across benchmarks measured in pass@1/pass@8.

| Model | AIME24 | AIME25 | AMC23 | GSM8k | MATH | Minerva | Olympiad | **Average** |
|---|---|---|---|---|---|---|---|---|
| | | | | Llama3.1-8B-Instruct | | | | |
| base | 1.7/10.9 | 0.4/2.8 | 15.0/47.3 | 57.7/92.8 | 29.3/55.9 | 14.7/38.5 | 2.2/8.2 | 17.3/36.6 |
| GRPO | 2.0/5.0 | 0.0/0.0 | 13.7/33.4 | 78.6/93.5 | 31.5/52.0 | 15.9/35.6 | 2.1/7.2 | 20.5/32.4 |
| $D^1S$ | 4.1/14.6 | **0.7/5.1** | 21.8/56.8 | 76.9/94.4 | **37.5**/59.6 | 19.9/41.9 | **3.8/11.6** | 23.5/**40.6** |
| $D^1S$-C | 2.7/9.3 | 0.1/0.8 | 14.4/35.7 | 78.4/93.4 | 31.3/51.7 | 16.2/35.5 | 1.9/7.0 | 20.7/33.3 |
| $D^2S$ | 1.9/6.5 | 0.1/1.0 | 13.1/32.3 | 77.6/93.5 | 32.3/53.0 | 15.9/36.8 | 2.3/7.7 | 20.5/33.0 |
| $D^3S$ | **5.3/20.7** | 0.1/0.8 | 20.3/50.8 | **79.0/95.0** | 35.9/59.2 | 22.5/44.3 | 3.3/10.7 | 23.8/40.2 |
| $D^3S$-I | 4.4/18.8 | 0.5/4.2 | **23.0/57.8** | 77.8/95.0 | 36.2/**59.6** | **22.8/44.6** | 3.3/10.6 | **24.0**/40.2 |
| | | | | OpenMath2-Llama3.1-8B | | | | |
| base | 3.3/12.2 | 2.0/10.2 | 35.5/60.7 | 89.1/96.2 | 49.8/65.6 | 11.8/26.4 | 7.8/15.0 | 28.5/40.9 |
| GRPO | 5.8/17.8 | 2.0/10.0 | 35.8/59.9 | 85.7/95.6 | 50.7/65.3 | 10.0/22.9 | 7.4/14.0 | 28.2/40.8 |
| $D^1S$ | **6.9/20.7** | **3.1/16.2** | **40.7/67.5** | **90.5**/96.1 | 52.6/66.2 | **14.0**/27.4 | 8.9/16.0 | **31.0/44.3** |
| $D^1S$-C | 6.8/20 | 2.5/11.6 | 35.9/63.6 | 89/95.9 | 52.2/**66.5** | 9.7/21.0 | 7.1/13.0 | 29.0/41.7 |
| $D^2S$ | 5.5/18.4 | 3/12.9 | 35.9/60 | 88.7/96 | 51.1/65.7 | 9.9/21.8 | 7.4/14.1 | 28.8/41.3 |
| $D^3S$ | 5.6/18.6 | 2.0/9.0 | 35.2/58.6 | 89.4/**96.2** | 49.6/64.9 | 11.3/26.3 | 8.1/15.2 | 28.7/41.3 |
| $D^3S$-I | 6.7/19.7 | 3.0/13.5 | 39.1/65.6 | 90.2/**96.2** | **52.7**/66.4 | 13.5/**27.5** | **9.0/16.4** | 30.6/43.6 |

between intra-group and cross-group sampling strategies narrows considerably. While the intra-group variants $D^1S$ (31.0/44.3) and $D^3S$-I (30.6/43.6) remain among the top performers, highlighting their robustness, the cross-group methods also yield competitive results. This suggests that as the model becomes better aligned and its response quality more consistent, the risk of instability from cross-group sampling diminishes, allowing its benefits to be more effectively realized.

This comparative analysis strongly indicates that the model's degree of alignment is a critical variable in determining the optimal sampling strategy. In this context, the $D^3S$-I variant stands out as a particularly robust framework. By combining the stability of intra-group sampling with the precision of token-level selection and dynamic scheduling, it delivers excellent performance and stable training dynamics across models with varying levels of initial capability, making it a more universally applicable solution.

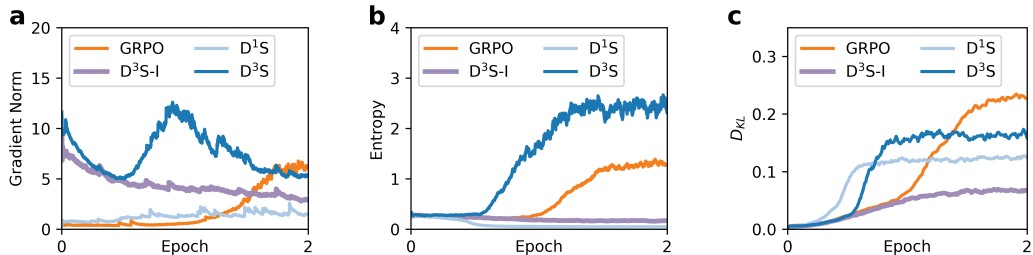

Figure 5: Detailed training dynamics of $D^3S$ strategies with Llama3.1-8B-Instruct as base model, where $D^3S$-I denotes $D^3S$ without cross-group down-sampling strategy. (a) gradient norms, (b) policy entropy, and (c) $D_{KL}$ restrain. Compared to the original GRPO, the integration of $D^3S$ significantly increases norm of policy gradient. Since Llama3.1-8B-Instruct model lacks pre-alignment, $D^3S$-I provides better balance in exploitation and exploration through smoother policy gradient and milder policy distribution migration measured in $D_{KL}$.

## D.2 COMPARISON WITH PODS

To address the specific inquiry regarding the performance difference between $D^3S$ and PODS (Xu et al., 2025), we provide a detailed comparison of their training dynamics.

**Sample Usefulness Rate (SUR).** As shown in Figure 6a, the SUR of PODS remains around 70% throughout the training process, exhibiting a pattern almost identical to the baseline GRPO. This indicates that PODS, which relies on intra-group selection, fails to utilize high-signal samples when they appear in "invalid groups" (groups where relative advantages are near zero). In contrast, $D^3S$ achieves a near 100% SUR in the early stages via cross-group sampling, ensuring maximum data efficiency when the model is weak.

**Training Stability** ($D_{KL}$). Figure 6b illustrates the KL divergence dynamics. PODS exhibits a significantly sharper rise in $D_{KL}$ compared to D³S, suggesting more aggressive and potentially unstable policy updates. D³S maintains a more controlled $D_{KL}$ trajectory, demonstrating that our dynamic scheduling effectively balances acceleration with training stability.

**Training Performance.** Figure 7 compares the Avg@32 accuracy on the validation set. While PODS provides a marginal acceleration over GRPO, D³S demonstrates a significantly steeper learning curve in the early stages and achieves a higher final convergent performance. This validates our theoretical analysis that maximizing Advantage Variance ($Var(A)$) yields a superior optimization landscape compared to maximizing Reward Variance ($Var(R)$) used in PODS.

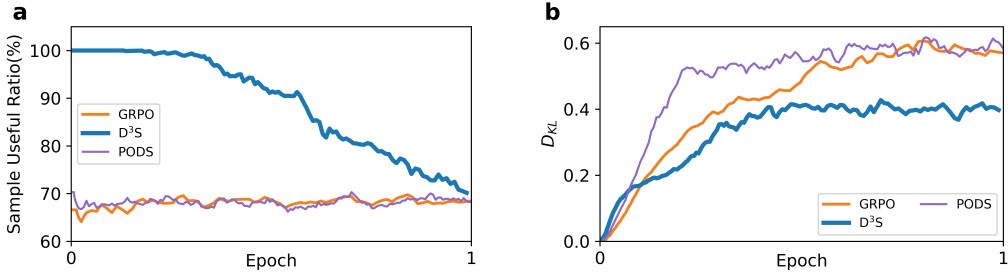

Figure 6: Training dynamics of (a) sample usefulness and (b) KL divergence compared among D³S, GRPO and PODS.

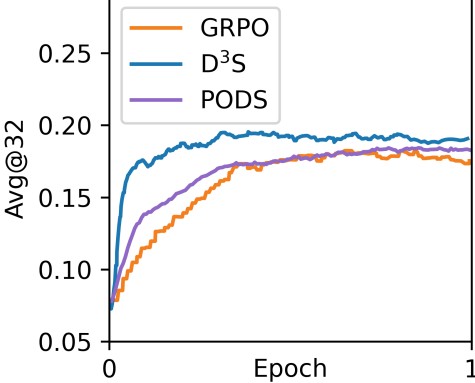

Figure 7: The Avg@32 test performance of AIME24 on Qwen2.5-Math-7B under various settings compared among D³S, GRPO and PODS.

### D.3 PASS@16 METRICS

Table 8: Performance across benchmarks measured in pass@16 calculated from 32 parallel runs.

| Model | AIME24 | AIME25 | AMC23 | GSM8k | MATH | Minerva | Olympiad | **Average** |
|---|---|---|---|---|---|---|---|---|
| | | | | Qwen2.5-Math-7B | | | | |
| base | 42.8 | 19.4 | 82.4 | 92.2 | 71.1 | 42.7 | 18.2 | 52.7 |
| GRPO | 42.7 | 28.0 | 89.0 | 96.3 | 73.1 | 50.2 | 22.3 | 57.4 |
| PODS | 48.1 | 30.8 | 85.1 | 96.4 | 73.5 | 51.9 | 23.2 | 58.4 |
| $D^1S$ | 52.3 | 27.4 | 89.9 | 96.5 | 73.2 | 52.7 | 22.8 | 59.3 |
| $D^1S$-C | 45.3 | **32.1** | 88.3 | 96.7 | 73.4 | 51.6 | 23.1 | 58.6 |
| $D^2S$ | 48.8 | 27.0 | 87.3 | 96.4 | 73.3 | 52.0 | 22.7 | 58.2 |
| **$D^3S$** | **54.6** | 31.6 | **91.2** | **96.9** | **73.9** | **53.4** | **23.3** | **60.7** |
| | | | | Llama3.1-8B-Instruct | | | | |
| base | 17.9 | 4.6 | 59.8 | 95.3 | 62.0 | 45.1 | 11.0 | 42.2 |
| GRPO | 6.7 | 0.0 | 44.7 | 94.9 | 57.2 | 41.2 | 9.7 | 36.3 |
| PODS | 14.7 | 5.0 | 51.0 | 94.9 | 57.8 | 44.5 | 10.0 | 39.7 |
| $D^1S$ | 18.5 | **8.8** | **70.0** | 95.6 | 64.7 | 47.4 | **14.6** | 45.7 |
| $D^1S$-C | 13.8 | 1.7 | 47.4 | 95.0 | 56.8 | 40.5 | 9.4 | 37.8 |
| $D^2S$ | 9.4 | 2.0 | 42.2 | 95.1 | 58.1 | 43.0 | 10.2 | 37.1 |
| **$D^3S$** | **28.4** | 1.7 | 59.9 | **96.3** | 64.7 | **49.9** | 13.6 | **44.9** |
| | | | | OpenMath2-Llama3.1-8B | | | | |
| base | 15.0 | 14.5 | 68.6 | **97.0** | 68.7 | 30.9 | 17.0 | 40.9 |
| GRPO | 22.5 | 13.7 | 65.2 | 96.4 | 68.2 | 27.2 | 15.9 | 44.2 |
| PODS | 22.6 | 17.7 | 67.9 | 96.7 | 68.3 | 25.8 | 16.2 | 45.0 |
| **$D^1S$** | **26.9** | **24.7** | **74.1** | 96.8 | **69.0** | 31.3 | **18.2** | **48.7** |
| $D^1S$-C | 24.2 | 15.7 | 70.2 | 96.8 | 69.6 | 24.5 | 14.6 | 45.1 |
| $D^2S$ | 25.9 | 18.1 | 67.6 | 96.7 | 68.8 | 25.8 | 15.9 | 45.5 |
| $D^3S$ | 24.0 | 13.2 | 65.3 | 96.9 | 68.1 | **31.3** | 17.0 | 45.1 |

