# OpenReview forum: "Learning More with Less: A Dynamic Dual-Level Down-Sampling Framework for Efficient Policy Optimization"
_ICLR.cc/2026/Conference — ICLR 2026 Poster_

### Official Review · Reviewer_CB3f · 2025-10-26

**Soundness:** 2
**Presentation:** 3
**Contribution:** 2
**Rating:** 4
**Confidence:** 2

**Summary:**

This study uncovers that in group-relative advantage-based RL algorithms (e.g., GRPO), the theoretical upper bound of the policy gradient norm scales with advantage variance. Building on this insight, the authors propose Dynamic Dual-level Down-Sampling (D3S), a novel training framework that improves efficiency and performance through two key mechanisms:

Sample-level: selects rollouts that maximize advantage variance to enrich signal diversity;
Token-level: prioritizes tokens with high entropy × |advantage|, focusing updates on regions that are both uncertain and influential.
To prevent overfitting, D3S employs a curriculum-inspired dynamic down-sampling schedule, gradually easing selection criteria as training progresses. Experiments on Qwen2.5 and Llama3.1 demonstrate consistent gains over baselines across multiple reasoning benchmarks.

**Strengths:**

(1) The thesis is well-written
(2) The theoretical proof is reasonable and supports the contribution of this paper
(3) The proposed algorithm is relatively concise and has certain generalization potential
(4) The experiment was conducted from multiple perspectives and achieved a good empirical analysis

**Weaknesses:**

Recently, there have been many works on trajectory screening based on entropy and advantage, and the methods are relatively homogeneous. Please distinguish the differences between this work and previous ones through detailed comparative experiments and discussions.

In my view, the amplitude of the advantage value represents the difficulty of the problem (like [3]), and the positive impact of high-entropy trajectories on the optimization of strategies has also been discussed in a large number of studies [1,2]. The combined influence of advantage and entropy was also discussed in the paper [4].

[1]: Beyond the 80/20 Rule: High-Entropy Minority Tokens Drive Effective Reinforcement Learning for LLM Reasoning
[2]: The Entropy Mechanism of Reinforcement Learning for Reasoning Language Models
[3]: Improving Data Efficiency for LLM Reinforcement Fine-tuning Through Difficulty-targeted Online Data Selection and Rollout Replay
[4]: RETHINKING ENTROPY INTERVENTIONS IN RLVR: AN ENTROPY CHANGE PERSPECTIVE

**Questions:**

Please refute the above weaknesses through discussion or experiment. I will adjust the score based on the author's feedback and the opinions of other reviewers.

---

> ### Author Response · Authors · 2025-11-20
> **Response to Reviewer CB3f**
>
> We sincerely appreciate your insightful comment and thank for highlighting these pivotal recent works. We have added a detailed discussion in **Section 5 (Related Works)**. While D$^3$S builds on the context of data sparsity[1] and entropy mechanisms[2][3] established by these studies, D$^3$S offers a distinct, systematic framework instead of a simple combination.
>
> We detail the systemtic design of D$^3$S below. We hope these analyses may address your concern.
>
> **1. Theoretical Foundation vs. Heuristic Difficulty (Sample-level)**
>
> Unlike DOTS [3], which relies on heuristic "difficulty"(based on pass rate) and auxiliary predictor models (adding computational cost), D$^3$S is grounded in Proposition 2 to maximize Advantage Variance ($Var(A)$) so as to directly maximizes the upper bound of the policy gradient norm. This data-driven, rollout-level approach adaptively captures fine-grained signals (e.g., a single high-variance error samples within a correct group. In DOTS, such groups are easy enough to ignore.) that prompt-level heuristics may miss.
>
> **2. Prioritization vs. Suppression (Token-level)**
>
> Prior works like Entropy Mechanism [2] and STEER [4] view high-impact tokens as sources of instability and propose suppressing them.
> D$^3$S takes the opposite approach. We identify high-signal tokens via the composite metric **$|A| \times H$** (filtering "useless uncertainty"). Instead of suppressing them, we prioritize them to accelerate convergence. Our experiments (Figure 3b-d) prove these tokens are assets, not risks, under scheduled prioritizing.
>
> **3. Solving the Stability-Efficiency Trade-off through Dynamic Scheduling**
>
> The "suppression" strategies[2][4] cited often sacrifice efficiency for stability. D$^3$S resolves this through the **Dynamic** Down-sampling Schedule. We aggressively exploit high-signal data early for speed (Efficiency), then gradually re-introduce lower-signal data to dilute gradients and stabilize training (Stability). This creates a cohesive framework that is effective across diverse algorithms (GRPO, GSPO) and models.
>
>
> In summary, **D$^3$S contributes an adaptive, systematic RLHF solution**. It theoretically grounds sample selection in variance maximization, refines token selection to target high-impact signals for acceleration, and employs dynamic scheduling to ensure stability. As demonstrated in our experiment, this framework is generic and effective across different algorithms (GRPO, GSPO) and base models (Qwen, Llama).
>
> ---
>
> [1]S. Wang *et al.*, “Beyond the 80/20 Rule: High-Entropy Minority Tokens Drive Effective Reinforcement Learning for LLM Reasoning,” June 02, 2025, *arXiv*: arXiv:2506.01939. doi: [10.48550/arXiv.2506.01939](https://doi.org/10.48550/arXiv.2506.01939).
>
> [2]G. Cui *et al.*, “The Entropy Mechanism of Reinforcement Learning for Reasoning Language Models,” May 28, 2025, *arXiv*: arXiv:2505.22617. doi: [10.48550/arXiv.2505.22617](https://doi.org/10.48550/arXiv.2505.22617).
>
> [3] Y. Sun *et al.*, “Improving Data Efficiency for LLM Reinforcement Fine-tuning Through Difficulty-targeted Online Data Selection and Rollout Replay,” [arXiv.org](http://arxiv.org/). Accessed: Nov. 14, 2025. [Online]. Available: https://arxiv.org/abs/2506.05316v3
>
> [4] Z. Hao *et al.*, “Rethinking Entropy Interventions in RLVR: An Entropy Change Perspective,” [arXiv.org](http://arxiv.org/). Accessed: Nov. 14, 2025. [Online]. Available: https://arxiv.org/abs/2510.10150v1

---

> > ### Comment · Reviewer_CB3f · 2025-11-25
> >
> > My concerns have been addressed, and I have increased the score to weakly accept.

---

> > > ### Author Response · Authors · 2025-11-26
> > > **Thanks to the re-evaluation and support of reviewer CB3f**
> > >
> > > We sincerely appreciate for your thoughtful feedback and for raising the score. We are delighted that our clarifications regarding the fundamental differences between D$^3$S and prior works (DOTS, STEER, etc.) have addressed your concerns. Discussion about metioned related researches helps strengthen our work and helps highlight unique design of D$^3$S.
> > >
> > > We will ensure that the detailed comparisons discussed are fully integrated into the final version of the paper.

---

### Official Review · Reviewer_Bn5B · 2025-10-29

**Soundness:** 4
**Presentation:** 4
**Contribution:** 4
**Rating:** 6
**Confidence:** 5

**Summary:**

This paper proposes a framework called Dynamic Dual-Level Down-Sampling ($D^{3}S$), aimed at making critic-free reinforcement learning algorithms such as GRPO more efficient. The core problem is that these methods are often slow to converge because critical learning signals are diluted by an abundance of uninformative samples and tokens.
Its core contributions are:
1) Theoretical Justification: It first theoretically proves that the upper bound of the policy gradient norm is positively correlated with the advantage variance ($Var(A)$) of the sampled subset. This justifies its strategy of selecting samples to maximize this advantage variance, rather than reward variance ($Var(R)$) as other methods do.

2) Dual-Level Down-Sampling: The $D^{3}S$ framework operates at the sample-level and token-level to select only the most valuable data for updates.

3) Dynamic Sampling Schedule: To prevent overfitting to this high-signal data, $D^{3}S$ employs a dynamic schedule inspired by curriculum learning. It starts with aggressive down-sampling (using fewer, high-signal samples) to accelerate early learning and then gradually relaxes this (using more data) to promote robust generalization.

Experiments on models like Qwen2.5 and Llama3.1 show that $D^{3}S$ achieves state-of-the-art performance on reasoning benchmarks while using significantly fewer tokens and achieving major training speedups.

**Strengths:**

1) This paper clearly defines the problem and presents a clear methodology, proposing a three-part strategy ($D^{3}S$) within critic-free frameworks like GRPO/GSPO to focus on the most informative samples and tokens from the source.
2) This paper provides sufficient theoretical proof, presenting the upper bound of the GRPO gradient norm (Proposition 1). It further proves that when maximizing advantage variance on a subset, this upper bound increases with the advantage variance (Proposition 2), which forms the theoretical basis for the sample-level selection. Subsequently, the dual selection levels are integrated into a PPO-style objective.
3) The experimental validation is thorough, conducted on models including Qwen2.5-Math-7B/1.5B and Llama3.1-8B, across a total of 7 benchmarks.

**Weaknesses:**

1) This paper introduces the Dynamic Down-Sampling Schedule, which in turn brings at least four key hyperparameters: $N_{init}$ (initial sample size), $N_{final}$ (final sample size), $K_{init}$ (initial token ratio), and $K_{final}$ (final token ratio). The paper demonstrates in Figure 3 that this dynamic schedule is crucial for preventing overfitting. This implies that the model's final performance may be highly dependent on the careful selection of these four new hyperparameters. However, the paper does not provide a sensitivity analysis for these hyperparameters. This could make the cost of reproduction and application to new tasks very high.

2) In Figure 4, $D^{3}S$ lowers policy entropy in well-aligned models (Qwen, OpenMath2) but sharply increases policy entropy in the unaligned Llama model. The paper attributes both of these contradictory behaviors to the ability of $D^{3}S$ to effectively balance exploration and exploitation. Why does the exact same algorithmic mechanism lead to exploration in one model and exploitation in another? The underlying mechanism for this is not fully substantiated.

**Questions:**

See weaknesses above.

---

> ### Author Response · Authors · 2025-11-20
> **Response to Reviewer Bn5B**
>
> We appreciate your valuable comments and try to address your concerns as follows.
>
>
> > Weakness 2: In Figure 4, D$^3$S lowers policy entropy in well-aligned models (Qwen, OpenMath2) but sharply increases policy entropy in the unaligned Llama model. The paper attributes both of these contradictory behaviors to the ability of D$^3$S to effectively balance exploration and exploitation. Why does the exact same algorithmic mechanism lead to exploration in one model and exploitation in another? The underlying mechanism for this is not fully substantiated.
> >
> We appreciate your insightful analysis and comments. Recent studies [1][2] on RLHF, along with their experimental results, have shown that entropy often decreases rapidly during the early stages of training and then steadily increases in later stages. Additionally, [3] has analyzed the relationship between increasing entropy and exploration. In our research, the observed entropy increase in **Figure 4c** serves as evidence of the adaptive exploration encouragement provided by the D$^3$S framework:
>
> - **For Llama3.1-8B-Instruct (Figure 4c):** This is a general-purpose model not aligned for mathematical reasoning. During fine-tuning, the baseline GRPO algorithm also leads to an increase in entropy. However, as shown by the GRPO curve, entropy remains low and stagnant for a longer period, indicating limited exploration efficiency. In contrast, D$^3$S, through its token-level selection mechanism (prioritizing tokens with high $|A_{i,t}| \times H_{i,t}$), encourages more effective and earlier exploration, enabling the model to escape local optima more efficiently. This is reflected in the observed rise in entropy.
>
> - **For OpenMath2 (Figure 4d):** This is a math-aligned model. Here, the baseline GRPO exhibits unstable high-entropy spikes, while D$^3$S successfully suppresses these fluctuations, maintaining low entropy to favor exploitation and stable convergence.
>
> In summary, D$^3$S does not arbitrarily increase or decrease entropy. Instead, it adapts to the model's state: promoting exploration (entropy increase) when the model is unaligned and ensuring stability (entropy decrease) when the model is already well-aligned.
>
> > Weakness 1: This paper introduces the Dynamic Down-Sampling Schedule, which in turn brings at least four key hyperparameters: $N_{init}$(initial sample size), $N_{final}$(final sample size), $K_{init}$(initial token ratio), and $K_{init}$(final token ratio). The paper demonstrates in Figure 3 that this dynamic schedule is crucial for preventing overfitting. This implies that the model's final performance may be highly dependent on the careful selection of these four new hyperparameters. However, the paper does not provide a sensitivity analysis for these hyperparameters. This could make the cost of reproduction and application to new tasks very high.
> >
> Many thanks for the insightful comments. Please refer to General Response for our detailed responses.
>
>
> We hope the analysis and revisions above may address your concerns strengthen our paper.
>
> ---
>
> [1]S. Wang *et al.*, “Beyond the 80/20 Rule: High-Entropy Minority Tokens Drive Effective Reinforcement Learning for LLM Reasoning,” June 02, 2025, *arXiv*: arXiv:2506.01939. doi: [10.48550/arXiv.2506.01939](https://doi.org/10.48550/arXiv.2506.01939).
>
> [2]G. Cui *et al.*, “The Entropy Mechanism of Reinforcement Learning for Reasoning Language Models,” May 28, 2025, *arXiv*: arXiv:2505.22617. doi: [10.48550/arXiv.2505.22617](https://doi.org/10.48550/arXiv.2505.22617).
>
> [3]D. Cheng *et al.*, “Reasoning with Exploration: An Entropy Perspective,” Nov. 08, 2025, *arXiv*: arXiv:2506.14758. doi: [10.48550/arXiv.2506.14758](https://doi.org/10.48550/arXiv.2506.14758).

---

> > ### Comment · Reviewer_Bn5B · 2025-11-26
> >
> > Thank you for the clarifications. My core concerns are addressed and I will keep my original score.

---

> ### Author Response · Authors · 2025-11-26
> **Thanks for confirmative feedback and support of reviewer Bn5B**
>
> We sincerely thank you for your time and effort spent re-evaluating our work. We are glad to hear that our rebuttal has successfully addressed your core concerns regarding hyperparameter sensitivity and the entropy mechanism.
>
> We will ensure that the additional analyses are incorporated into the final version of the paper. We appreciate your time and valuable feedback throughout this process.

---

### Official Review · Reviewer_wBN3 · 2025-11-01

**Soundness:** 3
**Presentation:** 2
**Contribution:** 2
**Rating:** 4
**Confidence:** 3

**Summary:**

The paper focuses on enhancing the learning efficiency of current LLM RL algorithms by training on subsets of rollouts with high advantage variance and tokens that exhibit both high advantage value and high entropy. Meanwhile, the algorithm is equipped with a dynamic schedule that includes more rollouts and tokens to prevent overfitting. An empirical comparison with GRPO, GSPO, and PODS is provided on multiple math reasoning benchmarks.

**Strengths:**

The algorithm is intuitive and well-explained.

The paper presents a thorough ablation study, demonstrating the contribution of each component.

**Weaknesses:**

PODS already proposed to downsample rollouts. The advantage of $D^3S$ over the previous model is not immediately apparent.

The analysis of token masking and dynamic scheduling is not thorough enough to understand their effects. (Check the Question section)

With the additional cost of hyperparameters, the performance improvement from PODS is not obvious.

The paper can be better prepared, for example, by adding a comparison to PODS for analysis and including y-axis labels for Figures 3 and

**Questions:**

1. How is the sample usage rate compared to PODS?
2. Is there any analysis on which tokens are masked off? Do they contain less information?
3. Is there any evidence that, without a downsampling schedule, the model is overfitting? As shown in Figure 2(a), the sample usage is low again. Why can such zero-advantage samples help if they do not modify the gradient?
4. The KL result where $D^3S$ is closer to the reference model is interesting. Why does it happen? Does it contribute to the performance of $D^3S$?
5. Usually, entropy decreases during RL training, but why does Figure 4 show even increasing entropies?

---

> ### Author Response · Authors · 2025-11-20
> **Response to Reviewer wBN3 - Part 1**
>
> Thank you for your thoughtful review and valuable feedback. Below, we discuss your concern about comparison between D$^3$S and a related research (PODS).
>
> > Weakness 1: PODS already proposed to downsample rollouts. The advantage of D$^3$S over the previous model is not immediately apparent.
> >
>
> We appreciate this critical insight. We clarify that the advantages of D$^3$S through aspects of theory, methodology, and empirical performance.
>
> 1.  **Theoretical Advantages:** The two methods optimize different objectives.
>
> - PODS optimizes **Reward Variance ($Var(R)$)**. As discussed in Section 2.1, this approach is ineffective for critic-free methods like GRPO, where normalized advantages are fixed ($Var(A) = 1$) and unaffected by $Var(R)$.
>
> - In contrast, D$^3$S optimizes **Advantage Variance ($Var(A)$)**. Proposition 2 demonstrates that this objective is theoretically superior for such algorithms, as it directly correlates with the upper bound of the policy gradient norm, enabling more effective optimization.
>
> 1.  **Methodological Advantages:**
>
> - PODS introduces a **second advantage estimation** on the selected subset, which can introduce bias due to recalculating advantages on a non-random, biased sample. This also increases computational overhead.  D$^3$S utilizes samples with original GRPO advantage signals group-relatively-estimated. So **D$^3$S  avoids potential re-estimation-bias**.
>
> - PODS suffers from the "invalid group" problem and relies on second advantage estimation to reconstruct the advantage signal. D$^3$S addresses this with **cross-group sampling**, which improves sample usefulness rate (SUR). As shown in new Appendix D.2 in the revision (For quick check: https://limewire.com/d/YeTG6#hwjTXY7W4T), PODS achieves a SUR of ~70% (similar to GRPO), indicating inefficiencies in sample batch remains. In contrast, D$^3$S maintains a SUR close to 100% (or dynamically controlled), ensuring as most as possible samples in a batch contributes effectively, particularly during the early-stage of training.
>
> 1. **Performance Advantages**:
> - We analyze the performance aspect in-detail in response to Weakness 3.
>
> > Weakness 3: With the additional cost of hyperparameters, the performance improvement from PODS is not obvious.
> >
>
> Thank you for your valuable feedback. We have conducted a thorough ablation study on hyperparameters, as outlined in the **General Response**. and Section 4.4 in revision paper. The results demonstrate that D$^3$S maintains robustness across the tested hyperparameter range, eliminating the need for meticulous tuning. For a detailed analysis, please refer to the General Response.
>
> Compared to PODS, the **empirical performance improvements** of D$^3$S can be summarized in three key aspects:
>
> - **Cost efficiency:** The $Var(A)$-driven sampling mechanism eliminates the computational overhead associated with additional advantage estimation required by PODS.
> - **Enhanced performance:** As shown in Table 1, D$^3$S consistently outperforms baseline methods, including GRPO, GSPO, and PODS.
> - **Improved generalizability:** Beyond its performance gains on Qwen2.5-Math-7B, D$^3$S demonstrates superior results on the **Llama3.1-8B-Instruct** model (Table 1). Without requiring additional hyperparameter tuning across different models, GRPO+D$^3$S achieves an average score of **23.8 / 40.2** (Pass@1/Pass@8), representing a significant improvement of **+3.1 (Pass@1) and +5.6 (Pass@8)** compared to GRPO+PODS (20.7 / 34.6).
>
> > Weakness 4: The paper can be better prepared, for example, by adding a comparison to PODS for analysis and including y-axis labels for Figures 3.
> >
>
> Thank your for the good suggestion. We have updated Figure 3d to include a comparison among GRPO, PODS, and D$^3$S, presented in Appendix D.2 in revision (For quick check: https://limewire.com/d/c6iiS#FZNf26IqhF). As shown in the figure, PODS demonstrates faster convergence during the early stages of training, but its improvement diminishes as training progresses. In contrast, D$^3$S not only accelerates early-stage convergence more effectively but also achieves a superior final convergence score.
>
> > Question 1: How is the sample usage rate compared to PODS?
> >
>
> We provide an improved comparison in Appendix D.2 in revision paper (For quick check: https://limewire.com/d/YeTG6#hwjTXY7W4T ). D$^3$S maintains a significantly higher Sample Usefulness Rate (SUR) than PODS by leveraging cross-group sampling to utilize high-signal samples from the entire batch.
>
> ---
>
> [1]Y. E. Xu, Y. Savani, F. Fang, and Z. Kolter, “Not All Rollouts are Useful: Down-Sampling Rollouts in LLM Reinforcement Learning,” June 06, 2025, *arXiv*: arXiv:2504.13818. doi: [10.48550/arXiv.2504.13818](https://doi.org/10.48550/arXiv.2504.13818).

---

> ### Author Response · Authors · 2025-11-20
> **Response to Reviewer wBN3 - Part 2**
>
> Thank you again for your thoughtful review and valuable feedback. In this part, we further discuss individual concerns about  D$^3$S. We hope below analysis may help address your concerns and strength this paper.
>
> > Weakness 2: The analysis of token masking and dynamic scheduling is not thorough enough to understand their effects.
> Question 2: Is there any analysis on which tokens are masked off? Do they contain less information?
> >
>
> Thank you for your thoughtful comments. Our analysis reveals that the tokens masked off are those with the lowest information value for policy improvement. As supported by prior research (e.g., the "80/20 rule"[1]), policy gradients in reasoning tasks are unevenly distributed, with a small subset (approximately 20%) of **high-entropy tokens** dominating the process. These tokens often represent critical logical transitions or turning points in mathematical reasoning. In D$^3$S, our token selection metric, $|A_{i,t}| \times H_{i,t}$, goes beyond simply identifying uncertain tokens (high $H$); it further prioritizes tokens based on their **impact on the final outcome** (high advantage magnitude, $|A|$). Consequently, D$^3$S masks off tokens that are either "easy" ($H \approx 0$, where the model is confident) or low-impact ($|A| \approx 0$, irrelevant to the outcome).
>
> The effectiveness of this token masking strategy is demonstrated by comparing the D$^2$S configuration (which incorporates token masking) with the D$^1$S variants (sample-level masking only). The benefits extend beyond final performance scores, encompassing efficiency and improved training dynamics:
>
> - **Efficiency:** As detailed in the ablation study in Section 4.3, D$^2$S achieves nearly identical final average performance (31.3/54.1) compared to D$^1$S (31.3/54.2), while significantly reducing the number of tokens processed during updates by focusing only on the top $K\%$ of informative tokens.
> - **Training Dynamics:** Token masking in D$^2$S enhances the training landscape. It accelerates early-stage convergence, as illustrated in **Figure 3(c)**, and maintains lower KL divergence in later training stages compared to D$^1$S variants, as shown in **Figure 2(b)**, indicating reduced distribution shift.
>
> In summary, token masking is a pivotal component that delivers substantial efficiency improvements and fosters more stable training dynamics.
>
> > Question 3: Is there any evidence that, without a downsampling schedule, the model is overfitting? As shown in Figure 2(a), the sample usage is low again. Why can such zero-advantage samples help if they do not modify the gradient?
> >
>
> We appreciate your valuable comments and try to address your concerns as follows.
>
> 1. **Evidence of Overfitting:**
> Figures 3(a)-(c) provide empirical evidence of overfitting when a downsampling schedule is absent. Static downsampling methods (D$^1$S and D$^2$S, represented in yellow and purple) show rapid initial performance gains but plateau or degrade in later stages, eventually being surpassed by the baseline GRPO. This decline, coupled with the larger KL divergence growth observed in Figure 2(b), indicates that static methods overfit to high-variance subsets and suffer from excessive distribution shifts.
> 2. **Role of Zero-Advantage Samples:**
> Zero-advantage samples contribute not by altering the gradient direction but by calibrating its magnitude. Through the dynamic schedule, these samples are gradually reintroduced, increasing the denominator in the batch average without significantly affecting the numerator. This effectively dilutes the gradient, acting as an implicit form of learning rate decay or regularization.

---

> ### Author Response · Authors · 2025-11-20
> **Response to Reviewer wBN3 - Part 3**
>
> Thank you again for your thoughtful review and valuable feedback. In this part, we further discuss individual concerns about  D$^3$S. We hope below analysis may help address your concerns and strength this paper.
>
> > Question 4: The KL result where D$^3$S is closer to the reference model is interesting. Why does it happen? Does it contribute to the performance of D$^3$S?
> >
>
> The KL result in **Figure 2(b)** is a direct outcome of our dynamic strategy.
>
> - **Early Training Phase:** In the initial stages of training (Figure 2b), the D$^3$S curve rises **steeper and faster** than the GRPO baseline. This demonstrates that by aggressively filtering out low-signal samples and focusing exclusively on high-advantage samples early on, D$^3$S delivers a cleaner and more robust learning signal. This enables the policy to align with the reward objective more rapidly than GRPO, resulting in the D$^3$S curve being above the GRPO curve.
> - **Later Training Phase:** As training progresses, the D$^3$S KL curve flattens and remains lower than those of static and baseline methods. These methods continue to focus on high-signal subsets, leading to higher KL increments. In contrast, D$^3$S incorporates **zero-advantage samples**, which influence the optimization landscape by diluting the gradient. Recent theoretical work on LLM loss landscapes, such as **FOCUS** [2], introduces the **"Narrowing Valley" hypothesis**. This hypothesis suggests that optimizers face a trade-off: they require large step sizes to navigate flat regions early in training but need smaller steps to converge into the sharper valley bottom later. The dynamic schedule of D$^3$S implicitly **aligns with this theoretical framework**, enabling it to achieve a better optimal position with reduced KL divergence shifts.
>
> > Question 5: Usually, entropy decreases during RL training, but why does Figure 4 show even increasing entropies?
> >
>
> Thank you for this insightful question. Recent studies [1][3] on RLHF have shown that entropy often decreases rapidly during the early stages of training but may steadily increase in later stages. In our research, the increasing entropy observed in Figure 4c highlights the adaptive nature of the D$^3$S framework. Specifically, Llama3.1-8B-Instruct, being a general-purpose model not aligned for mathematical reasoning, exhibits entropy increases even under the baseline GRPO algorithm during fine-tuning. However, as shown in the GRPO curve, entropy remains low and stagnant for a prolonged period, indicating limited exploration efficiency.
>
> In contrast, D$^3$S leverages its token-level selection mechanism (prioritizing tokens with high $|A_{i,t}| \times H_{i,t}$) to promote more effective and earlier exploration. This allows the model to escape local optima more efficiently, as reflected in the observed rise in entropy. For comparison, Figure 4d (OpenMath2, a math-aligned model) shows that while the baseline GRPO produces unstable high-entropy spikes, D$^3$S successfully suppresses this volatility, maintaining low entropy to favor exploitation and stable convergence.
>
> In summary, D$^3$S does not rigidly increase or decrease entropy. Instead, it adapts to the model's state: encouraging exploration (entropy increase) when the model is unaligned and promoting stability (entropy decrease) when the model is already capable.
>
> ---
>
> [1]S. Wang *et al.*, “Beyond the 80/20 Rule: High-Entropy Minority Tokens Drive Effective Reinforcement Learning for LLM Reasoning,” June 02, 2025, *arXiv*: arXiv:2506.01939. doi: [10.48550/arXiv.2506.01939](https://doi.org/10.48550/arXiv.2506.01939).
>
> [2]Y. Liu, Z. Liu, and J. Gore, “FOCUS: First Order Concentrated Updating Scheme,” Jan. 21, 2025, *arXiv*: arXiv:2501.12243. doi: [10.48550/arXiv.2501.12243](https://doi.org/10.48550/arXiv.2501.12243).
>
> [3]G. Cui *et al.*, “The Entropy Mechanism of Reinforcement Learning for Reasoning Language Models,” May 28, 2025, *arXiv*: arXiv:2505.22617. doi: [10.48550/arXiv.2505.22617](https://doi.org/10.48550/arXiv.2505.22617).

---

> > ### Comment · Reviewer_wBN3 · 2025-11-25
> > **Questions Clarified**
> >
> > Thanks for your thorough explanation! My questions are well answered, and I have increased the score to weakly accept.

---

> > > ### Author Response · Authors · 2025-11-25
> > > **Thank you for the re-evaluation and support of reviewer wBN3**
> > >
> > > ​We sincerely thank you for the positive feedback and for raising the score. We are glad that our responses and the new ablation studies have resolved your concerns.
> > >
> > > ​We will ensure that all the clarifications are incorporated into the final version of the paper. Your constructive comments have significantly strengthened our work.

---

### Official Review · Reviewer_5NNR · 2025-11-11

**Soundness:** 3
**Presentation:** 3
**Contribution:** 3
**Rating:** 8
**Confidence:** 2

**Summary:**

This paper proposes a down-sampling approach for fine tuning LLMs with RL to speed up learning and convergence called D3S. The authors argue that using all the available interaction data can slow down learning because critical learning signals are overshadowed by many uninformative transitions. The proposal is to down-sample training data at two levels: (1) select a small set of trajectories to have large advantage variance, and (2) within each trajectory select transitions whose advantage magnitude multiplied by policy entropy is large. Finally, to avoid overfitting to the sub-selected data, the sub-sampling is decreased over training to promote large updates in early learning, while maintaining diversity and generalization toward the end of training.

The contributions are as follows,
1. Proposing a down-sampling framework to improve training efficiency of critic-free policy optimization algorithms for fine tuning LLMs.
2. Theoretical analysis to prove that maximizing the advantage variance increases the upper bound of the policy gradient norm, which can speed up learning.
3. Experiments show improved sample efficiency, and better final performance on benchmarks compared with previous methods
4. Ablation studies show different components of the algorithm contribute to its improved performance

**Strengths:**

- I believe this paper is a good contribution to the RL and LLM literature and should be considered for acceptance. Down-sampling training data to improve training efficiency is not common in prior work and may be interesting to the community. Furthermore, maximizing advantage by sub-selecting trajectories is a novel idea and it is nicely investigated in this work.

- The paper's overall quality is good. The writing and content organization is good. The exposition and motivation is clear. The paper mostly does a good job covering prior work and placing their contribution in the context of the broader literature. The proposed algorithm is supported by intuitive and theoretical justification and the experiments are accompanied by sufficient details and analysis.

**Weaknesses:**

- The paper may benefit from further discussion of prior work such as coresets for machine learning. For example: Mirzasoleiman, B., Bilmes, J.A., & Leskovec, J. (2019). Coresets for Data-efficient Training of Machine Learning Models. International Conference on Machine Learning.
- The paper may also benefit from some discussion of prioritizing transitions in RL. For example: Schaul, T., Quan, J., Antonoglou, I., & Silver, D. (2015). Prioritized Experience Replay. arXiv: Learning.

- some typos in the abstract and introduction
  - line 017 “efficient”
  - line 019 “proven”
  - line 078 “data pool is gradually expanded”

**Questions:**

- The paper claims state of the art generalization for D3S. I don't think this claim is supported by the evidence. Maybe this claim needs further elaboration on what is meant by generalization in this case.
- D3S also introduces some hyperparameters such as N_init, K_init, N_final, K_final. Are these values hard to choose or require additional resources to find good values for? Are they sensitive to the task?

---

> ### Author Response · Authors · 2025-11-20
> **Response to Reviewer 5NNR**
>
> Thank you for your valuable feedback and appreciate your recognition of our work's advancements. Below, we address your concerns in detail.
>
> > Questions 1: The paper claims state of the art generalization for D$^3$S . I don't think this claim is supported by the evidence. Maybe this claim needs further elaboration on what is meant by generalization in this case.
> >
>
> We appreciate you raising this crucial point. In our context, "generalization" refers to :
>
> 1. **Versatility across setups:** The capability of D$^3$S to improve RLHF performance across different base algorithms (GRPO/GSPO), model families (Qwen2.5/Llama 3.1), scales (1.5B/7B), and alignment stages (Llama 3.1 Base/OpenMath Llama), as well as across diverse reasoning benchmarks (AIME, GSM8K, etc.), without requiring extensive task-specific hyper-parameter tuning.
> 2. **Robustness across data distributions:** The capability of D$^3$S to sustain performance improvements across the full data distribution, rather than overfitting to high-signal subsets selected early in training. As illustrated in Section 4.5 and Figure 3, static downsampling accelerates early convergence but often leads to overfitting on high-variance samples, causing subsequent performance degradation. D$^3$S addresses this via a curriculum learning-inspired dynamic mechanism. It prioritizes high-signal data initially to accelerate convergence, then gradually relaxes sampling criteria to incorporate diverse samples. This ensures the model effectively leverages high-signal data while maintaining the robustness required to generalize across the broader distribution.
>
> To ensure precision, we have revised the term "state-of-the-art performance and generalization" in the abstract and clarified the description of generalization in Section 3.3 of the revised version.
>
> > Questions 2: D$^3$S also introduces some hyperparameters such as N_init, K_init, N_final, K_final. Are these values hard to choose or require additional resources to find good values for? Are they sensitive to the task?
> >
>
> Many thanks for the insightful comments. Please refer to **General Response** for our detailed responses.
>
> > Weakness 1: The paper may benefit from further discussion of prior work such as coresets for machine learning. For example: Mirzasoleiman, B., Bilmes, J.A., & Leskovec, J. (2019). Coresets for Data-efficient Training of Machine Learning Models. International Conference on Machine Learning.
> >
> > Weakness 2: The paper may also benefit from some discussion of prioritizing transitions in RL. For example: Schaul, T., Quan, J., Antonoglou, I., & Silver, D. (2015). Prioritized Experience Replay. arXiv: Learning.
> >
>
> We appreciate your recommendation of related researches which share similar design philosophy with our study. Below, we provide a more detailed discussion:
>
> Coresets[1] aim to select a weighted subset of data that closely approximates the full gradient during supervised learning. In contrast, D$^3$S does not attempt to approximate the average gradient of the entire batch. Instead, it discards samples with insufficient information (i.e., those with advantages close to 0) and selects a "core subset" that maximizes the advantage variance ($Var(A)$). Our objective is to create a steeper optimization landscape to accelerate convergence.
>
> Prioritized Experience Replay (PER)[2] uses the TD-error ($|\delta|$) as a priority metric to replay transitions where the model's predictions are less accurate more frequently. D$^3$S, however, operates on two levels: (1) at the sample level, it maximizes the Advantage Variance ($Var(A)$); and (2) at the token level, it leverages the Advantage Magnitude ($|A_{i,t}| \times H_{i,t}$). While PER is an off-policy method, its underlying principle of prioritizing high-signal data rather than suppressing it aligns with the design philosophy of D$^3$S.
>
> We have revised Section 5 (Related Works) to incorporate the above discussion.
>
> > Weakness 3: some typos in the abstract and introduction
> >
>
> Thank you for your meticulous review. We have corrected the typos in the revised version and will carefully proofread the final version before submission.
>
> ---
>
> [1] B. Mirzasoleiman, J. Bilmes, and J. Leskovec, “Coresets for Data-efficient Training of Machine Learning Models,” Nov. 16, 2020, arXiv: arXiv:1906.01827. doi: 10.48550/arXiv.1906.01827.
>
> [2] T. Schaul, J. Quan, I. Antonoglou, and D. Silver, “Prioritized Experience Replay,” Feb. 25, 2016, arXiv: arXiv:1511.05952. doi: 10.48550/arXiv.1511.05952.

---

### Author Response · Authors · 2025-11-20
**General Response**

We sincerely thank all reviewers for their thoughtful and constructive feedback, which are crucial for improving our work.

**A key concern** raised commonly (by reviewers 5NNR, Bn5B, and wBN3) is the sensitivity of the dynamic scheduling hyperparameters ($N$ and $K$). In response to it, we conducted a thorough ablation study, and the corresponding results have been included in the revised manuscript (see Section 4.4 and Table 4). These results demonstrate that D$^3$S remains robust to variations in hyperparameters within the tested range and consistently outperforms the vanilla GRPO.

- **Sample Size ($N$):** Whether using $N = 8 \rightarrow 16$ or $N = 16 \rightarrow 32$, the average performance (Avg Pass@1/Pass@8) aligns closely with our original setting ($N = 8 \rightarrow 32$) in the paper. $N = 16 \rightarrow 32$ achieves an Avg Pass@1/Pass@8 of 34.41/56.16, closely matching the original configuration's 34.84/56.76.

- **Token Ratio ($K$):** We evaluated various $K$ configurations, and the Pass@8 performance consistently falls within a narrow range (54.20% to 56.76%). Interestingly, a broader range configuration ($K = 5\\% \rightarrow 30\\%$) achieved a higher Pass@1 performance (36.34%), further validating our scheduling strategy of progressively increasing data diversity.

These ablation studies confirm that the effectiveness of D$^3$S is not tied to specific hyperparameter settings or complex tuning. The sampling strategy of D$^3$S inherently balances high-signal data with diversity. By adopting a straightforward and intuitive dynamic range (e.g., starting with smaller $N,K$ values and scaling up), significant performance improvements can be consistently achieved.

**In response to other individual concerns**, we have also made detailed replies and necessary revisions in paper corrspondingly. We believe these revisions may address the reviewers' concerns and significantly strengthen the paper.

In favor of reviewers’ ease to check in discussion, ablation results of hyper-parameters are presented below. Baseline configuration of D$^3$S is $N=8\rightarrow 32, K=5\\%\rightarrow 20\\%$ with rollout number of GRPO as $G=32$.

| **Configuration** | **AIME24** | **AIME25** | **AMC23** | **GSM8k** | **MATH** | **Minerva** | **Olympiad** | **Average** |
| --- | --- | --- | --- | --- | --- | --- | --- | --- |
| GRPO | 13.2/37.6 | 5.5/21.6 | 47.0/83.5 | 64.9/94.3 | 48.5/70.2 | 19.8/45.0 | 9.7/19.8 | 29.8/53.1 |
| **D$^3$S** | **20.3**/**48.2** | 7.9/25.8 | 54.4/**87.1** | 73.4/95.7 | 52.2/71.5 | **25.0**/**48.2** | 10.7/20.8 | 34.3/**56.8** |
| $N=8\rightarrow 16$ | 20.1/46.2 | 7.9/23.2 | 56.5/85.0 | 76.1/**95.9** | 55.5/**71.7** | 23.0/46.9 | 11.1/21.0 | 35.8/55.7 |
| $N=16\rightarrow 32$ | 18.4/46.7 | 7.6/25.2 | 55.0/85.5 | 73.6/95.8 | 53.1/71.4 | 22.5/47.6 | 10.7/20.9 | 34.4/56.2 |
| $K=5\\%\rightarrow 30\\%$ | 19.7/43.4 | **8.9**/25.6 | **57.4**/81.3 | **77.6**/95.7 | **55.9**/**71.7** | 23.6/47.6 | **11.3**/21.2 | **36.3**/55.2 |
| $K=5\\%\rightarrow 40\\%$ | 18.1/41.1 | 7.6/24.6 | 55.2/82.5 | 72.5/95.4 | 54.1/71.5 | 21.7/46.9 | 10.2/20.8 | 34.2/54.7 |
| $K=5\\%\rightarrow 50\\%$ | 17.5/44.5 | 7.6/25.8 | 52.0/82.8 | 74.4/95.6 | 54.2/**71.7** | 24.1/47.9 | **11.3**/**21.4** | 34.4/55.7 |
| $K=10\\%\rightarrow 20\\%$ | 16.8/42.3 | 7.5/23.8 | 56.0/83.6 | 74.3/95.6 | 54.6/71.6 | 21.7/46.4 | 10.5/20.5 | 34.5/54.8 |
| $K=10\\%\rightarrow 30\\%$ | 17.3/46.1 | 8.1/24.8 | 53.7/86.4 | 72.7/95.2 | 53.7/71.5 | 22.1/47.0 | 10.3/20.8 | 34.0/56.0 |
| $K=10\\%\rightarrow 40\\%$ | 18.6/45.0 | 7.7/26.1 | 52.5/83.2 | 70.2/95.6 | 52.2/71.4 | 22.5/**48.2** | 10.4/20.6 | 33.4/55.7 |
| $K=10\\%\rightarrow 50\\%$ | 17.9/44.5 | 8.0/**27.8** | 52.3/83.6 | 72.7/95.6 | 53.9/71.5 | 22.2/47.2 | 10.7/20.7 | 34.0/55.8 |

---

> ### Comment · Area_Chair_KRwE · 2025-11-21
> **Author-Reviewer Discussion**
>
> Dear reviewers,
>
> Please review the authors' response and adjust your rating accordingly. If you have any further questions, please discuss with the authors further.
>
> AC

---

### Author Response · Authors · 2025-12-02
**General Response - Part 2**

# Dear Area Chair and Reviewers,

We understand the unprecedented challenges caused by recent data leak bug and subsequent rollback. We sincerely appereciate all the time, comments and support from all Chairs and Reviewers, which have all helped us improve our research.

To assist your assessment, we summarize the actual consensus reached during the discussion period.

Prior to the rollback, we have successfully addressed all concerns from reviewers (before Nov 26).

**All reviewers who responsed explicitly confirmed that their concerns were resolved. In short:**

- **Reviewer 5NNR:** **Score 8 (Accept)**. Evaluated as *"excellent paper" initially*.
- **Reviewer Bn5B:** **Score 6** . Confirmed: *"My core concerns are addressed."*
- **Reviewer wBN3:** Raised Score **4 $\to$ 6** . Stated: *"My questions are well answered, and I have increased the score to weakly accept."*
- **Reviewer CB3f:** Raised Score **4 $\to$ 6**  Stated: *"My concerns have been addressed, and I have increased the score to weakly accept."*

We hereby present a brief summary of our resolution of concerns:

## 1. Hyper-parameter ablation

**Reviewers 5NNR, Bn5B and wBN3** commonly expressed one concerns regarding the sensitivity of the newly introduced hyperparameters ($N_{init}, N_{final}, K_{init}, K_{final}$) for the dynamic schedule.

**To address this**, we conducted a comprehensive ablation study reported in **General Response** and **Section 4.4 in revision paper**. Results confirmed that D$^3$S is robust and does not require extensive hyperparameter tuning.

Reviewers confirmed that this ablation has addressed their concerns.

## 2. Learning dynamics

**Reviewers Bn5B and wBN3** questioned the seemingly contradictory entropy behavior (entropy increase in Llama vs. decrease in others) and the mechanism behind the lower KL divergence and the role of "zero-advantage" samples.

**To address this**, we provided a deeper analysis of the training dynamics:

- **Entropy Mechanism:** We clarified such behavior is evidence supporting that D$^3$S is **adaptive**. On unaligned models (Llama, Fig 4c), it drives necessary **productive exploration** (leading to more entropy increase than baseline) by targeting tokens where the model is confident but wrong. On aligned models (OpenMath2, Fig 4d), it suppresses volatility (leading to smoother, lower entropy curve than baseline) to **favor exploitation**.
- **KL & Stability:** We explained that the **Dynamic Schedule** acts as an implicit gradient stabilizer. By re-introducing lower-advantage samples in the later phase, D$^3$S dilutes the average gradient norm. This prevents the policy from overfitting to high-variance samples, resulting in a lower and more stable KL divergence (Fig 2b) compared to static methods.

We have improved the **Section 4.7 and Appendix.D.2 in revision paper** to better illustrate these discussion.

Reviewers confirmed that these analysis and revision has addressed their concerns.

## 3. Design Philosophy

**Reviewers wBN3 and CB3f** questioned the novelty of D$^3$S compared to prior works like PODS , STEER , and DOTS , requiring further analysis about them.

D$^3$S offers a distinct, systematic framework to resolve the "Efficiency-Stability" trade-off in RLHF. **To address reviewers’ concern**, we articulated below fundamental differences in our **Theoretical Objective** and **Optimization Philosophy**:

- **Theory (vs. PODS):** To improve RLHF efficiency, D$^3$S maximizes **Advantage Variance ($Var(A)$)**, which **Proposition 2** proves correlates with a **higher gradient norm upper bound** in GRPO. We clarified such fundamental theoretical difference with PODS, which maximizes *Reward Variance* ($Var(R)$), which we showed is theoretically ineffective for changing the gradient bound. Furthermore, D$^3$S employs **cross-group selection** to avoid advantage re-estimation bias and "invalid group" issues inherent in PODS's intra-group selection.
- **Method & Philosophy (vs. STEER/Entropy Mechanism):** We highlighted a diametrically opposed philosophy. While prior works view high-signal tokens as risks for stability and **suppress** them, D$^3$S views them as assets and **prioritizes** them for efficiency. We solve the resulting stability trade-off not by suppression, but by the **Dynamic Schedule**.

We have improved the **Related Works (Section 5) and Appendix.D.2 in revision paper** to include expanded discussion about these researches.

Reviewers confirmed that these clarifications and revision have answered their questions.

## 4. misc

We improved and summitted a thourough revision, with including discussions, correcting typos and clarifying the "generalization" concept mentioned by **Reviewer 5NNR** specifically. We appreciate for all review comments which have helped a lot to strengthen our research.

Best regards,

The Authors

---

### Meta-Review · Area_Chair_ExKU · 2026-01-06

**Summary:**

Across reviewers, the paper’s core idea—dynamic dual-level down-sampling for critic-free RLHF-style optimization (trajectory selection by advantage variance + token selection by |A|×entropy, with a relax-to-full-data schedule)—is viewed as intuitive and potentially impactful for speeding convergence and improving final performance. One reviewer is strongly positive, and two borderline reviewers moved to weak accept after rebuttal clarifications and added analyses.
The remaining decision-relevant issues were mainly about positioning vs. closely related downsampling/prioritization work (PODS/PER/entropy-based screening), the validity of the “generalization/SOTA generalization” claim, the new hyperparameters introduced by the schedule, and whether the paper provides a convincing mechanism-level explanation for observations like KL/entropy dynamics.

**Reviewer Concerns:**

Most major concerns appear addressed in the rebuttal: the authors clarified what they mean by “generalization” (cross-model/algorithm/benchmark robustness + avoiding overfitting via the dynamic schedule) and agreed to soften phrasing in the abstract/sections. They also added explicit discussion/comparisons to PODS (objective mismatch for critic-free GRPO, bias from re-estimating advantages, and improved sample usefulness rate), plus qualitative explanations and ablations for token masking and the schedule; notably, the PODS-focused reviewer stated their questions were answered and raised the score to weak accept.
What remains outstanding is largely camera-ready execution: the paper must clearly and fairly differentiate from the growing set of entropy/advantage screening methods with concrete comparative evidence in the main text, and it should include at least a minimal sensitivity/robustness story for the schedule hyperparameters (N_init/K_init/N_final/K_final) rather than pointing to a “general response.” The entropy explanation is plausible but still risks reading as post-hoc unless tied tightly to measurable signals and consistent plots (labels, units, and definitions).

**Reviewer Scores:**

Given the discussion, 5NNR is firmly in accept territory and would likely keep an accept score (their main asks were literature positioning and clarifying the “generalization” claim). Reviewer wBN3 explicitly increased to weak accept after the rebuttal, and reviewer CB3f also increased to weak accept once differences to prior works were clarified.
Reviewer Bn5B stayed at marginal accept despite initially raising reproducibility concerns; after rebuttal they said the core concerns were addressed but did not upgrade. Net-net, the panel shifts from mixed to mostly accept/weak accept, with remaining risk being presentation and reproducibility clarity rather than technical soundness.

---

### Decision · Program_Chairs · 2026-01-26

Accept (Poster)